# A cell fitness selection model for neuronal survival during development

Yiqiao Wang [1,11], Haohao Wu[1,11], Paula Fontanet[1], Simone Codeluppi[2], Natalia Akkuratova[3], Charles Petitpré[1], Yongtao Xue-Franzén[1], Karen Niederreither[4], Anil Sharma [1], Fabio Da Silva[5], Glenda Comai [6], Gulistan Agirman[1], Domenico Palumberi[1], Sten Linnarsson [2], Igor Adameyko[3,7], Aziz Moqrich[8], Andreas Schedl[5], Gioele La Manno[9], Saida Hadjab[1,12] & François Lallemend[1,10,12]

Developmental cell death plays an important role in the construction of functional neural circuits. In vertebrates, the canonical view proposes a selection of the surviving neurons through stochastic competition for target-derived neurotrophic signals, implying an equal potential for neurons to compete. Here we show an alternative cell fitness selection of neurons that is defined by a specific neuronal heterogeneity code. Proprioceptive sensory neurons that will undergo cell death and those that will survive exhibit different molecular signatures that are regulated by retinoic acid and transcription factors, and are independent of the target and neurotrophins. These molecular features are genetically encoded, representing two distinct subgroups of neurons with contrasted functional maturation states and survival outcome. Thus, in this model, a heterogeneous code of intrinsic cell fitness in neighboring neurons provides differential competitive advantage resulting in the selection of cells with higher capacity to survive and functionally integrate into neural networks.

[1] Department of Neuroscience, Karolinska Institutet, 17177 Stockholm, Sweden. [2] Unit of Molecular Neurobiology, Department of Medical Biochemistry and Biophysics, Karolinska Institutet, 17177 Stockholm, Sweden. [3] Department of Physiology and Pharmacology, Karolinska Institutet, Stockholm 17165, Sweden. [4] Institut de Génétique et de Biologie Moléculaire et Cellulaire (IGBMC), CNRS UMR7104, Inserm U964, Université de Strasbourg, Illkirch, France. [5] Université Côte d'Azur, Inserm, CNRS, iBV, 06108 Nice, France. [6] Stem Cells & Development - Institut Pasteur - CNRS UMR3738, 75015 Paris, France. [7] Center for Brain Research, Medical University Vienna, Vienna, Austria. [8] Aix-Marseille-Université, CNRS, Institut de Biologie du Développement de Marseille (IBDM), UMR 7288, 13288 Marseille, France. [9] Brain Mind Institute, School of Life Sciences, Swiss Federal Institute of Technology (EPFL), Lausanne, Switzerland. [10] Ming Wai Lau Centre for Reparative Medicine, Stockholm node, Karolinska Institutet, Stockholm, Sweden. [11] These authors contributed equally: Yiqiao Wang, Haohao Wu. [12] These authors jointly supervised this work: Saida Hadjab, François Lallemend. Correspondence and requests for materials should be addressed to S.H. (email: saida.hadjab@ki.se) or to F.L. (email: francois.lallemend@ki.se)

Programmed cell death plays a fundamental role in animal development and tissue homeostasis. In the mammalian nervous system, massive cell death and axonal degeneration represent an essential process in the formation of functional neural circuits during development[1,2]. The accepted mechanism driving the death of neurons during this critical period involves a stochastic competition between neurons for limiting access to target-derived neurotrophic factors[3,4]. Yet, how or which given cells are chosen to survive or die is still poorly understood.

In the vertebrate nervous system, the majority of neurons form during embryonic development when, after a phase of neurogenesis, a vast number of them are quickly removed by apoptosis in order to optimize quantitatively the connectivity between neurons and their targets (systems matching) and control the final architecture of the neuronal structures[1,3,4]. In the peripheral nervous system, survival of selected neurons depends on target-derived neurotrophins (NTs)[5,6], which bind to specific tropomyosin receptor kinase (TRK) receptors on the neuronal cell surface[7]. Typically, the binding of TRK receptors by their cognate NTs at the nerve endings induces a retrograde transport signaling that supports the survival, growth and differentiation of the neurons[8,9]. The neurotrophic theory is the prevailing model for the regulation of neuronal cell death during development[3,4]. This model proposes that neurons produced in excess compete for a limited supply of NTs synthesized in their target fields[4,10–12]. Built into this model is also the assumption that the probability of individual neurons to survive the competition for NTs cannot be predicted[10,13,14], implying that all neurons are endowed with an equal potential to compete. However, very early on, neurons exhibit intrinsic differences that control their differentiation and axon targeting[15–17], emphasizing the need to re-examine the stochastic nature of the neurotrophic theory and to determine whether cell-to-cell molecular variability and survival predictability represent strong selection mechanisms during neuronal competition for NTs-induced survival (Fig. 1a). In this context, a more deterministic life-or-death regulation, coupled with these intrinsic differentiation mechanisms, would provide a way for selecting the fittest cells during development.

Here, we tested this hypothesis using tropomyosin receptor kinase C (TRKC)[+] proprioceptive sensory neurons (PSNs) of the dorsal root ganglia (DRG) and demonstrate that the survival probability is not equal amongst neurons and hence some neurons are more likely to die. We show that before cell death period starts, the presumptive PSNs display different molecular blueprints initiated by the morphogen retinoic acid (RA) and which lead through transcriptional regulation to distinct subgroups of neurons with different functional fitness and survival outcome. Thus, our data demonstrate a predisposition to cell death during development, where molecular fitness heterogeneity in sensory neurons predicts their survival probability and the selection of the fittest cells for the establishment of functional neural networks.

## Results

**Heterogeneity of TRKC expression in PSNs before cell death.** To investigate whether neurons are equivalent prior to the cell death period, we used the DRG that have been extensively studied for neuronal cell death regulation during development. We focused on the development of the PSNs, which later innervate skeletal muscles and represent a major neuronal population of the DRG of the trunk[18]. During embryogenesis, PSNs are characterized by the expression of the transcription factors RUNX3 and of TRKC, the specific receptor for neurotrophin-3 (NT3) (Fig. 1b)[17–19]. At embryonic day (E) 11.5, most TRKC[+] DRG neurons at brachial levels are RUNX3[+] (and all RUNX3[+] neurons are TRKC[+]), the vast majority of them being actual PSNs, as

assessed by genetic tracing of TRKC-expressing neurons from E11.5 to E17.5 using *TrkC[CreER];R26[tdTOM]* mice (Fig. 1c and Supplementary Fig. 1)[18,20–22]. On the other hand, NT3 is synthetized in the mesenchyme and the early muscle mass surrounding the growing axons of TRKC[+] PSNs during development[23]. Upon binding to TRKC, NT3 retrogradely activates pro-survival signaling events that are required for PSNs to survive the cell death period, which occurs from E12.0 to E12.5 and before target innervation (Fig. 1d)[24–26].

The classic neurotrophic hypothesis considers that neurons have equal capacity to compete and thus, to survive the developmental cell death period, implying similarity in their molecular features before they reach their target[13,14]. However, at E11.0–11.5, analysis of the molecular identity of PSNs showed a high variability in their TRKC expression, with no obvious spatial pattern or a correlation with the cell size (Fig. 1e–i and Supplementary Fig. 2). This heterogeneity was maintained in vitro, with TRKC levels correlating between the soma and nerve endings where NT3 engages its receptor in vivo (Supplementary Fig. 3). The functional, full length (FL) tyrosine kinase isoform of *Ntrk3*[7] was the predominant isoform ($64.34 \pm 1.03\%$) in E11.5 brachial DRG when analyzed by real-time quantitative polymerase chain reaction (PCR), and results from immunostainings and single-molecule RNA fluorescence in situ hybridization (smFISH) confirmed the high cell-to-cell variable expression of the *Ntrk3* FL isoform (Fig. 1j, k and Supplementary Fig. 4). Thus, our results clearly indicate that prior to cell death period, TRKC FL expression in individual PSNs is highly heterogeneous.

To examine whether the different levels of TRKC in PSNs arise from protein accumulation during neurogenesis (from E9.5 to E10.5 in mice)[27], with the early born neurons having accumulated more protein over time than later born counterparts, we fate-mapped early born TRKC neurons. For this, we induced recombination in *TrkC[CreER];R26[tdTOM]* mice at ~E9.75 with a single injection of 4-hydroxytamoxifen (4-OHT, 0.06 g/kg). Convincingly, recombination in PSNs was not correlated with their TRKC levels observed at E11.5 as tdTomato (TOM)[+] cells analyzed at this stage were similarly distributed among the TRKC[High] and TRKC[Low] categories of PSNs (Fig. 1l–n), indicating that TRKC heterogeneity is independent of birthdate.

**TRKC levels are associated with competitive advantage.** The high variability of TRKC levels among PSNs at early developmental stages suggests a target-independent control of their molecular heterogeneity. Here, we explored the possible influence of the environment on TRKC levels, focusing on NT3, which is expressed around the projecting axons in the limb[23]. Using real-time quantitative PCR and immunohistochemistry, we showed that TRKC expression is completely independent of NT3 both in vitro and in vivo (Fig. 2a–c); *Etv1* expression, known to depend on NT3 in vivo[28], showed, however, a 4.4-fold increase in NT3 condition ($P < 0.001$, unpaired Student's *t*-test). In line with that hypothesis, a previous work in embryonic chicken DRG showed that ablation of the limb does not alter TRKC expression in the corresponding DRG neurons prior to cell death, indicating that TRKC levels in PSNs are established independently of target-derived signals[17]. To study the transcriptional control of TRKC levels, we focused on the transcription factor RUNX3, which is known to transactivate the *Ntrk3* locus[29] (*Ntrk3* codes for TRKC), and showed a tight correlation with TRKC levels in E11.5 PSNs (Fig. 2d). Analysis of *Runx3[−/−]* mice at E11.5 showed significantly lower levels of TRKC in PSNs (now similar to the TRKC[Low] category in Ctr animals), with however similar number of neurons (Fig. 2e, f). This confirmed earlier studies on the role of RUNX3 on TRKC levels but not on its induction[17,19]. In

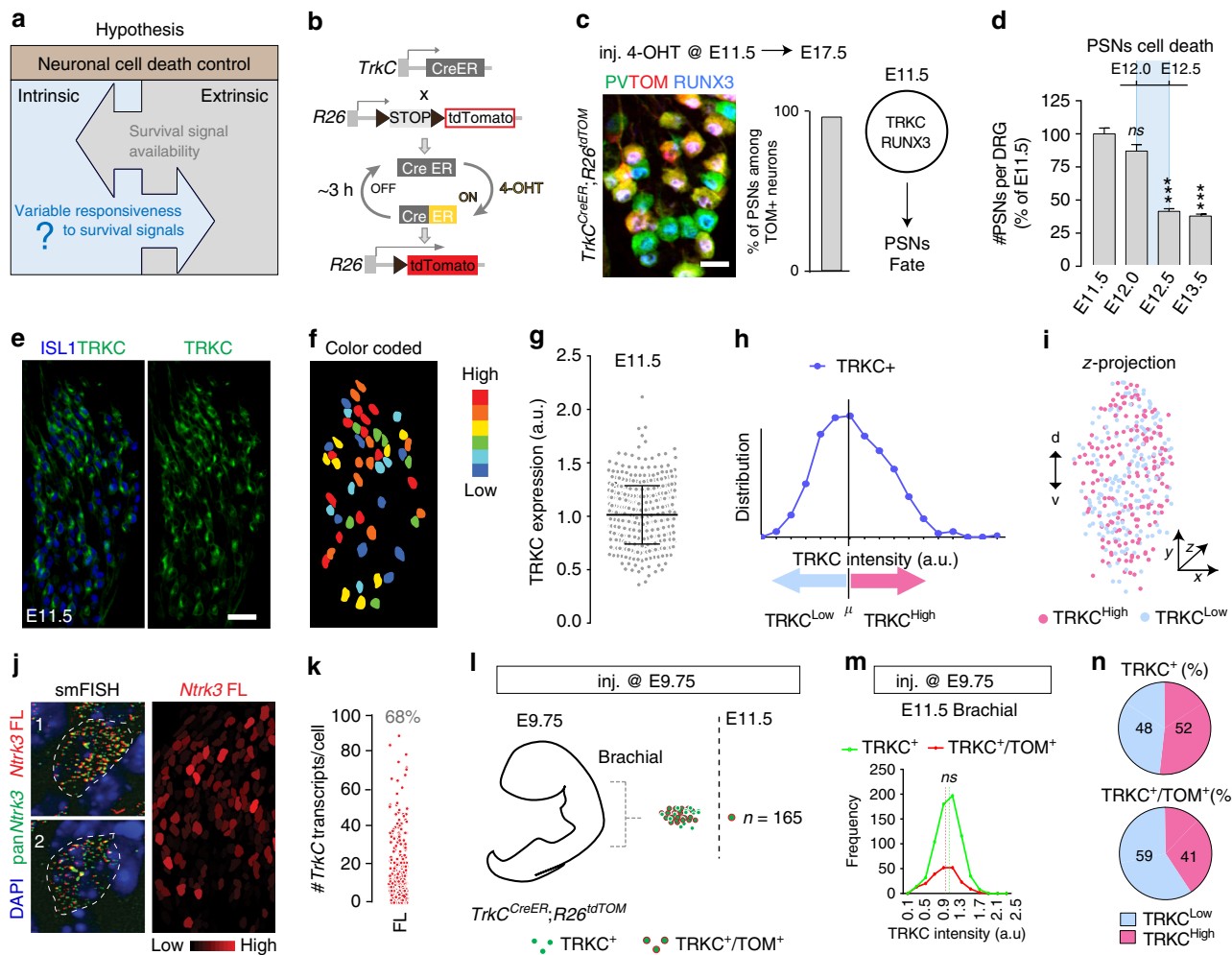

**Fig. 1** Differential expression of TRKC in PSNs prior to the cell death period. **a** Scheme of our working hypothesis. **b**, **c** Temporal fate mapping of TRKC PSNs by 4-OHT induction. *TrkC^CreER* mice allow temporary activation of CreER in the TRKC+ cells 2 h after 4-OHT injection[21,22]. Immunostaining for PV, RFP and RUNX3 on E17.5 DRG sections (**c**) and graph showing distribution of PV+/RUNX3+ PSNs among the TOM+ cells (*n* = 4). Scale bar: 20 μm. **d** Quantification of PSNs at C5 and C7. ***P < 0.001, one-way analysis of variance (ANOVA) with Sidak's multiple comparisons test (*n* = 2–3). The window of PSNs cell death is shown. **e** TRKC expression in E11.5 ISL1+ (and RUNX3+, whose staining is not shown for more visibility) DRG neurons. Scale bar: 50 μm. **f** TRKC levels in PSNs of **e** illustrated by color coding; dark blue indicates the lower and red the higher TRKC levels. From here, all observations are done at brachial levels (C5–8). **g** Distribution of TRKC levels in PSNs from **e**. **h** Distribution of TRKC levels in PSNs in E11.5 DRG neurons (from **g**). The data exhibit a Poisson-like distribution (one representative animal), with the mean used to define the two different categories of TRKC intensity (TRKC^High and TRKC^Low). **i** Projection of seven images of RUNX3+/TRKC+ PSNs from one brachial DRG; dots indicate TRKC-labeled neurons and color codes reveal TRKC intensity as shown in **h**. **j** Projection image of smFISH for pan *Ntrk3* and *Ntrk3* full length (FL) transcripts in E11.5 DRG, visualized at high magnification in (1) and (2) (images show full projection); right panel shows color coding of *Ntrk3* FL levels in red; the brighter, the higher levels. **k** Distribution of the number of *Ntrk3* FL molecules in E11.5 DRG neurons by smFISH, normalized to pan *Ntrk3* (*Ntrk3* FL represent 68% of all *Ntrk3* transcripts). **l** *TrkC^CreER*; *R26^tdTOM* mice were injected at E9.75 with 4-OHT and analyzed at E11.5 (*n* = 3). **m**, **n** Frequency distribution (**m**) and pie chart (**n**) of TOM+/TRKC+ neurons from **l** according to their level of TRKC intensity. Source data are available as a Source Data file

contrast, RUNX3 levels remained unchanged up to E12.5 in the absence of NT3-TRKC signaling (Supplementary Fig. 5), confirming that NT3 does not induce or regulate RUNX3 expression[19]. Collectively, our data support a model in which the variable levels of TRKC expression in PSNs are intrinsically regulated prior to cell death period in a target- and NT-independent manner.

Given the essential role of TRKC signaling in the survival of developing PSNs, we sought to analyze the possible relationship between TRKC levels in PSNs and their survival capacity. We found a positive correlation between the levels of TrkC and the expression of the phosphorylated form of the serine/threonine-specific protein kinase AKT, which acts within the TRKC signaling survival pathway during the cell death period[7] (Fig. 2g),

suggesting more survival advantage for neurons with high relative TRKC levels. For our hypothesis to hold, lower TRKC-expressing PSNs would be more vulnerable to limited amount of NT3. We thus tested whether the levels of TRKC in PSNs impact their responsiveness to NT3 and hence their survival capacity. We took advantage of *Runx3^−/−* mice, which have very low levels of TRKC in PSNs and in which all PSNs die at E12.5 (Fig. 2e, f)[17,30]. We challenged PSNs from *Runx3^−/−* DRG in vitro with different concentrations of NT3. Compared to wild-type neurons, PSNs from *Runx3^−/−* mice did not respond to NT3-dependent survival at low concentration (1 ng/ml, EC50, Supplementary Fig. 6b) while a saturating concentration of NT3 (50 ng/ml) could promote survival, although to a lesser extent than in wild-type (Fig. 2h). This indicates that in the *Runx3^−/−* animals (with low

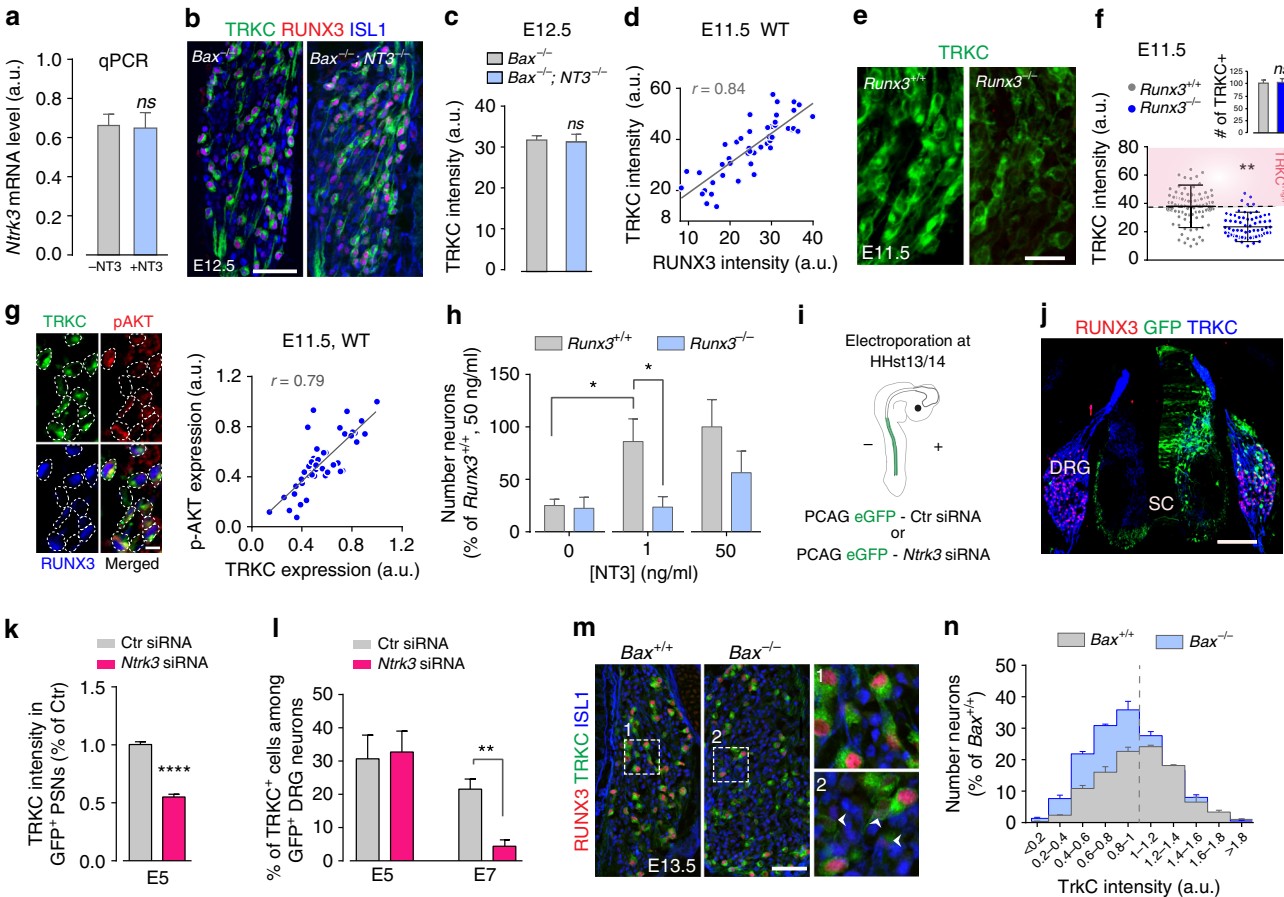

**Fig. 2** Regulation of TRKC levels in PSNs and their correlation with differential survival. **a** Quantification of *Ntrk3* in E11.5 DRG cultured for 6 h with or without NT3 (50 ng/ml) (*n* = 4). **b** TRKC/RUNX3 immunostaining on E12.5 DRG sections from *Bax*⁻/⁻ and *Bax*⁻/⁻;*NT3*⁻/⁻ animals[34]. Scale bar: 50 μm. **c** Quantification of TRKC levels in individual PSNs from E12.5 *Bax*⁻/⁻ and *Bax*⁻/⁻;*NT3*⁻/⁻ animals in **b** (*n* = 2), with similar number of PSNs in both *Bax*⁻/⁻ and *Bax*⁻/⁻;*NT3*⁻/⁻ DRG (*P* = 0.83, unpaired Student's *t*-test). **d** Correlation between RUNX3 and TRKC levels in individual E11.5 PSNs. **e** TRKC expression in E11.5 DRG sections from *Runx3*⁺/⁺ and *Runx3*⁻/⁻ animals. Scale bar: 20 μm. **f** Quantification of TRKC intensity in individual PSNs in E11.5 *Runx3*⁺/⁺ and *Runx3*⁻/⁻ animals. **P* < 0.01. Bar graph: number of TRKC⁺ neurons in E11.5 *Runx3*⁺/⁺ and *Runx3*⁻/⁻ animals; ns, not significant, unpaired Student's *t*-test (*n* = 3). **g** Immunostaining for TRKC, RUNX3 and p-AKT on E11.5 DRG sections and quantified correlation between TRKC and p-AKT levels in individual PSNs (right panel). **h** Survival of E11.5 DRG neurons from *Runx3*⁺/⁺ and *Runx3*⁻/⁻ animals treated with NT3 for 48 h (**P* < 0.05, *n* = 4–6, one-way analysis of variance (ANOVA) with Sidak's multiple comparisons test). **i** Scheme representing the in ovo electroporation (HHst 13/14, or E2). **j** Cross section of electroporated HHst26/27 embryos (E5) showing transfected cells in half spinal cord (SC) and in the ipsilateral DRG. **k** Quantification at E5 (before cell death period) of TRKC intensity among transfected PSNs (RUNX3⁺/TRKC⁺). ****P < 0.0001, unpaired Student's *t*-test. **l** Comparison before (E5) and after cell death period (E7) of the % of PSNs in transfected GFP⁺ neurons. **P < 0.01, unpaired Student's *t*-test (*n* = 4). **m** TRKC and RUNX3 expression in PSNs of E13.5 *Bax*⁺/⁺ and *Bax*⁻/⁻ DRG sections. (Right) Boxed regions (1) and (2) are magnified. Arrowheads show PSNs with lower size (see Supplementary Fig. 6c). Scale bar: 50 μm. **n** Quantification of TRKC levels in PSNs from E13.5 *Bax*⁺/⁺ and *Bax*⁻/⁻ DRG neurons (*n* = 4). Dashed line distinguishes TRKC^High and TRKC^Low populations. Source data are available as a Source Data file

TRKC levels in PSNs), the NT3-TRKC survival signaling is still responsive but only to high concentrations of NT3. To further support a role for TRKC levels in predisposition of neurons to apoptosis during competition for NTs, we set out to reduce but not abolish the level of TRKC in PSNs by electroporating chick embryos at Hamburger and Hamilton stage 13 (HHst13; before neural crest migration, around E2) with siRNA either directed against *Ntrk3* and a GFP plasmid (*Ntrk3* siRNA) or with a negative control siRNA and GFP plasmid (*Ctr* siRNA) (Fig. 2i). This strategy allows fate tracing of neural crest-derived *Ntrk3* siRNA-expressing PSNs, which exhibit low levels of TRKC (Fig. 2j). It also enables the targeting of only a small proportion of the neurons (~7% of PSNs), hence avoiding possible indirect role on neurogenesis as previously shown in mice and chicken embryos in the absence of NT3-TRKC signaling[23,31,32]. Analysis

of DRG sections at HHst26/27 (E5), i.e. before the cell death period of PSNs[33], revealed no noticeable loss of GFP⁺ PSNs in *Ntrk3* siRNA condition (Fig. 2k). In contrast, at HHst30/31, after the cell death period of PSNs, there was an 80% decrease in the number of GFP⁺ PSNs electroporated with *Ntrk3* siRNA, when compared with DRG sections from embryos transfected with *Ctr* siRNA (Fig. 2l). This suggests that low TRKC-expressing PSNs are less fitted to compete for NTs during the cell death period and predict a greater survival of neurons with higher TRKC levels. Inhibition of survival competition by means of cell death blockade using *Bax*⁻/⁻ mice (in which all sensory neurons survive)[34] should then prevent this selection and result in a proportional increase of neurons with low TRKC. In *Bax*⁻/⁻ DRG, the number of PSNs at E13.5 is about 2.2-fold higher (TRKC⁺/RUNX3⁺ neurons from C5–C8; wild-type (WT):

$100 \pm 13.7$, $Bax^{-/-}$: $228 \pm 12.6.\%$ of WT), reflecting a rate of cell survival that approximates 50% in wild-type PSNs. Consistent with our prediction, this increase was accompanied by an overrepresentation of neurons with low TRKC expression (Fig. 2m, n), a large proportion of them being atrophied (Supplementary Fig. 6c), suggesting that prevention of natural PSNs cell death results in the accumulation of neurons with relative lower TRKC content, which would have been eliminated in a wild-type context. Together, these data indicate a strong, positive relationship between TRKC levels, trophic signaling state and the capacity to compete and survive.

**Differential survival of PSN subpopulations in vivo.** To directly assess the higher survival probability of PSNs with high TRKC levels in vivo, we developed a Cre recombinase-dependent methodology to genetically trace PSNs as a function of TRKC expression. Indeed, by using the CreER driver and a low dose of 4-OHT, the fraction of cells undergoing recombination should depend on the amount of Cre recombinase they express[35]. To test this in vivo, we induced limited recombination in $TrkC^{CreER}$; $R26^{tdTOM}$ mice at E11.0 with a single injection of 4-OHT (0.018 g/kg) and analyzed the traced cells before (E11.5) and after the cell death period (E13.5) (Fig. 3a). At E11.5, recombination was observed in 11% of PSNs, predominantly labeling neurons

with high TRKC levels (with 75% of TOM$^+$ PSNs expressing levels of TRKC above the average) (Fig. 3b–d), supporting our strategy to increase the probability of tracing high TRKC$^+$/high Cre$^+$ neurons by limiting the Cre activity. Importantly, this preferential tracing of the high TRKC neurons also demonstrate that the difference in TRKC levels in PSNs is relatively stable during this time period. Strikingly, when assessing the survival of the labeled PSNs after the cell death period (E13.5), we found an 1.88-fold increase in the proportion of labeled PSNs (from 11.6% at E11.5 to 20.9% at E13.5; Fig. 3e, f), confirming our prediction and supporting our hypothesis. In sharp contrast, injection of $TrkC^{CreER}$;$R26^{tdTOM}$ mice at E14.0 (after the cell death period of PSNs) with a similar dose of 4-OHT led to an identical proportion of recombined PSNs at both E14.5 (13.2%) and E16.5 (13.5%) (Fig. 3g). Moreover, similar experiments using an $R26^{CreERT2}$ driver line, which labels PSNs independently of their TRKC levels, did not show any enrichment in the proportion of recombined PSNs (TOM$^+$ amongst TRKC$^+$/RUNX3$^+$ neurons) between E11.5 and E13.5 (Fig. 3g and Supplementary Fig. 7).

Altogether, our data demonstrate that neuronal competition selects for neurons with higher TRKC levels during development. Since NT3 expression in the limb does not seem limited to particular sub-regions of PSNs targets at E11.5[23], it is highly unlikely that this selection mechanism could reflect the possibility that PSNs with differential levels of TRKC would project

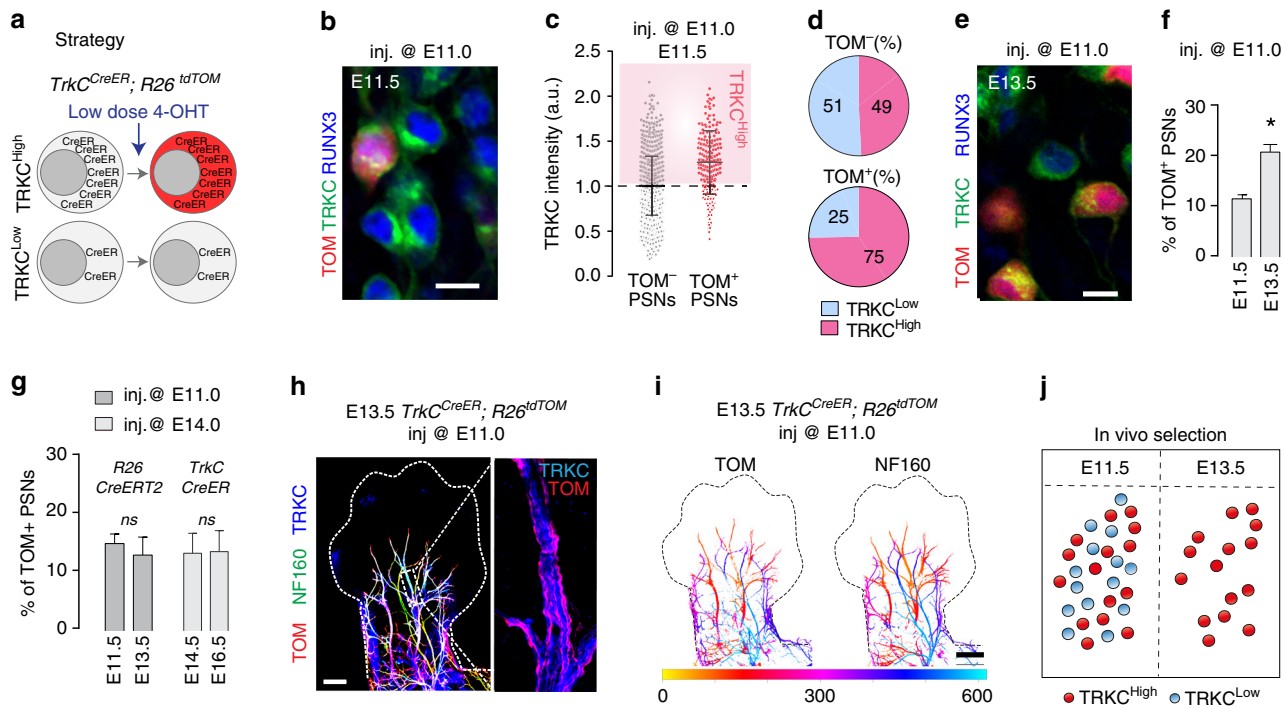

**Fig. 3** PSNs with high TRKC levels preferentially survive the cell death period. **a** Temporal fate mapping of TRKC$^{High}$ PSNs by 4-OHT (low dose, 0.02 g/kg). **b–d** Injection of $TrkC^{CreER}$;$R26^{tdTOM}$ mice with low dose of 4-OHT at E11.0; DRG analyzed at E11.5 with recombination in few (**b**), preferentially high TRKC PSNs (**c, d**) ($P < 0.001$). Frequency distribution of TOM$^+$ PSNs according to TRKC intensity (**c**) and pie charts (**d**) illustrating the large proportion of TOM$^+$ cells among TRKC$^{High}$ PSNs. Scale bar: 50 μm. **e, f** Percentage of recombined PSNs at E11.5 and E13.5 in DRGs from $TrkC^{CreER}$;$R26^{tdTOM}$ animals after 4-OHT injection at E11.0 (*$P < 0.05$, Student's t-test; $n = 2$ litters with 6 embryos, E11.5; 2 litters with 5 embryos, E13.5). **g** The percentage of labeled PSNs does not change between E11.5 and E13.5 in $R26^{CreERT2}$;$R26^{tdTOM}$ embryos injected at E11.0 with 0.032 g/kg 4-OHT ($n = 4$). Similarly, the recombination rate in TOM$^+$ PSNs does not change between E14.5 and E16.5 in $TrkC^{CreER}$;$R26^{tdTOM}$ embryos after injection at E14.0 with 0.02 g/kg 4-OHT ($n = 2$). Unpaired Student's t-test. **h** Whole-mount immunostaining for TRKC, NF160 and RFP of E13.5 forelimb from $TrkC^{CreER}$;$R26^{tdTOM}$ embryos injected with low dose 4-OHT at E11.0. Insert shows restricted number of TOM$^+$ fibers dispersed amongst TRKC$^+$ axons. Scale bar: 200 μm. **i** Pattern and color-coded depth (in micrometers) of innervation of TRKC$^+$, NF160$^+$ and RFP$^+$ nerve fibers (processed from **h**) of E13.5 forelimb from $TrkC^{CreER}$;$R26^{tdTOM}$ embryos injected with a low dose of 4-OHT (0.02 g/kg) at E11. The pattern and depth color code reveal similar territories (in all dimensions, $xyz$) of innervation of the TOM$^+$ PSNs compared to all axons (NF160). Scale bar: 200 μm. **j** Scheme illustrating the preferential selection of TRKC$^{High}$ PSNs during the cell death period. Source data are available as a Source Data file

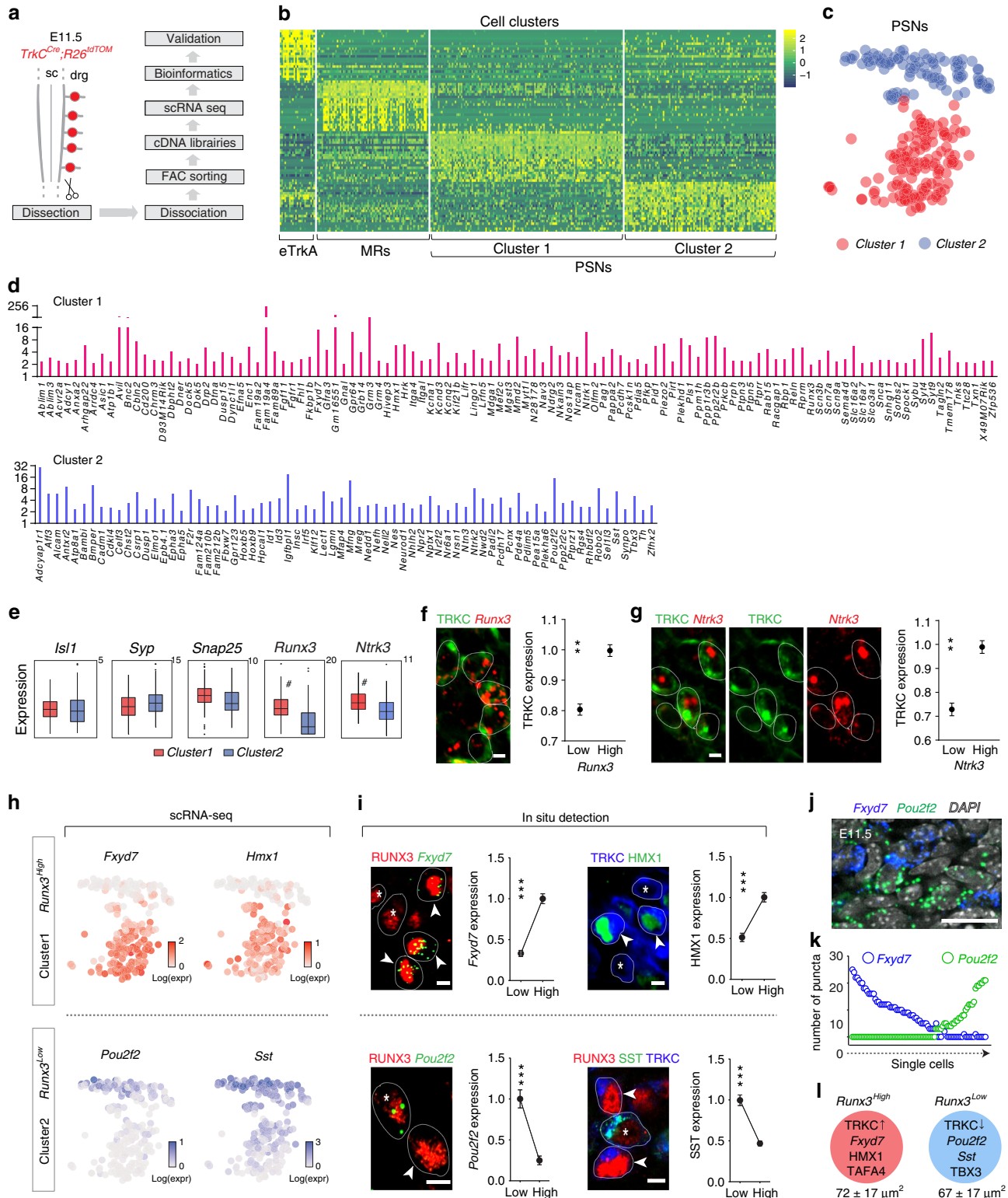

peripherally to distinct target fields, with distinct availability to NTs. In support of this, defasciculation of sensory projections within the limb in mutant mice for axon guidance molecules has no impact on the neuronal cell death process[36,37], and peripheral projections of PSNs with high TRKC levels did not show any preference of spatial distribution within peripheral nerves (Fig. 3h, i). Altogether, this strongly suggests a selection model in which neighboring neurons with differential fitness compete for NTs in

the target tissue, resulting in the elimination of cells with lower advantage to compete and survive (Fig. 3j).

**Cell fitness characters of PSN subpopulations.** Our findings of the survival selection of one PSNs group (expressing high levels of TRKC and of RUNX3) over the other group suggest that these two groups might represent two genetically distinct PSNs

**Fig. 4** Single-cell RNA-seq identifies two molecularly distinct populations of PSNs prior to cell death period. **a** Workflow for single-cell RNA-seq (scRNA-seq) processing of DRG neurons. **b** Heat map showing single-cell expression of the top 20 differentially expressed genes in the different clusters of DRG neurons, identifying Clusters 1 and 2 as PSNs. **c** *t*-SNE plot showing two distinct populations of *Ntrk3+/Runx3+* PSNs at E11.5, the Clusters 1 and 2 in **b**. eTRKA: early TRKA neurons; MR: mechanoreceptive neurons[18]; both populations transiently express TRKC at an earlier time point[38]. **d** List of Cluster 1- (top panel) and Cluster 2-enriched (bottom) genes. **e** The two clusters express similar levels of classic neuronal markers but differ in their level of *Runx3* and *Ntrk3* transcripts and are named *Runx3High* and *Runx3Low*. Box plots visualize the summary of the dataset (minimum, lower quartile, median, upper quartile and maximum). Wilcoxon rank sum test (# indicates significant difference, see Methods section for details). **f, g** In vivo validation of the correlation between TRKC protein and *Runx3* mRNA (**f**) and between TRKC protein and *Ntrk3* mRNA (**g**). ***P* < 0.01, unpaired Student's *t*-test (*n* = 3). **h–j** Marker genes of the *Runx3Low* and *Runx3High* subgroups of PSNs; *t*-SNE plots (**h**) and in vivo validation by immunostaining or in situ hybridization (**i**). Graphs in **i** show quantification of the expression of marker genes for *Runx3High* (High) and *Runx3Low* (Low) PSNs (*n* = 3 animals for each experiment), expressed in average number of dots for transcripts and average immunostaining intensity for proteins (and normalized to the maximum average value). ****P* < 0.001, unpaired Student's *t*-test. Scale bar: 5 μm. **j** In situ hybridization on E11.5 DRG sections showing the (almost) mutually exclusive expression of *Fxyd7* and *Pou2f2*. Scale bar: 15 μm. **k** Quantitative illustration of **j**, where for each *TrkC+* neurons, their level of *Fxyd7* and *Pou2f2* is represented. **l** Schematic representation of neuronal types with their key markers and their average soma size (in square micrometers). Source data are available as a Source Data file

subgroups. To test this hypothesis, a total of 350 TOM+ cells from *TrkCCre;R26TOM* E11.5 DRG, in which the majority of the neurons generated between E9.5 and E10.5 are traced[38], were processed for single-cell transcriptome analysis (Fig. 4a). The cell expression data from *Ntrk3+/Runx3+* PSNs were clustered using R package SEURAT and visualized using bi-dimensional *t*-distributed stochastic neighbor embedding (*t*-SNE) (Fig. 4b, c). Interestingly, two clusters of PSNs were identified unbiasedly, and were associated with distinct levels of *Runx3*: a *Runx3High* group and a *Runx3Low* group (Fig. 4b–e and Supplementary Data 1). It is important to note that both subgroups are PSNs since their genetic tracing at E11.5 lead at E17.5 to 95% of labeled cells with a PSN phenotype (Fig. 1c and Supplementary Fig. 1) and also because they all can develop into PSNs when NT3 is overexpressed in developing muscles[39]. These two PSNs subgroups were further distinguished based on their differential expression of marker genes (Fig. 4d). *Runx3High* group was characterized by the expression of *Hmx1*, *Fxyd7*, *Fam19a4* and an enrichment in *Ntrk3* while the *Runx3Low* group expressed *Sst*, *Pou2f2* and *Tbx3* (Fig. 4d, h). The correlation between *Runx3* mRNA level and TRKC protein, and between TRKC protein and *Ntrk3* mRNA, was then validated in vivo (Fig. 4f, g), as well as the existence of the two PSNs subgroups (Fig. 4h–k). Our results thus confirmed the existence of two PSNs subgroups based on their gene expression and that may correspond to the two PSNs groups with distinct survival outcome (Fig. 4l). We explored further our dataset and performed a gene ontology analysis to reveal benchmark biological categories behind the genetic blueprint of the *Runx3High* and *Runx3Low*. The top significant categories shared by both subgroups were biological categories associated with neuron development, axogenesis and neuron projection (Fig. 5a–c and Supplementary Fig. 8a). The two subgroups also showed similar levels of expression of genes involved in energy/metabolism (Supplementary Fig. 8b). Though, the two subgroups were embodied by distinct categories that underlined their transcriptome differences (Fig. 5d). Thus, the *Runx3High* group showed enrichment in neurotransmission and synaptic signaling categories (Fig. 5d, e), which could underlie a higher maturation of the functional properties of this subgroup. Notably, the levels of expression of the voltage-gated sodium channels *Scn3b* ($Na_v\beta3$), *Scn7a* ($Na_x$) and *Scn9a* ($Na_v1.7$), which contribute to the generation of a mature action potential[40], and of *Piezo2*, the main mechanotransduction channels in non-noxious mechanosensitive sensory neurons[41,42], were much higher in the *Runx3High* subgroup (Fig. 5f). To visualize the direction of developmental change within our population of cells, we applied RNA velocity to our single-cell measurements (PSNs from E11.5 and E12.5) to predict transcriptional dynamics and the cell state trajectory of

our neurons[43]. We found similar clustering at E11.5, with the *Runx3Low* to the *Runx3High* subgroups, but only one homogeneous cluster of PSNs at E12.5. The RNA velocity analysis showed a strong directional flow from the *Runx3Low* to the *Runx3High* subgroups and then to the E12.5 stage (Fig. 5h–k), confirming that the subgroups of PSNs observed at E11.5 represent neurons of various cell states along their maturation trajectory. In support, marker genes of *Runx3Low* are not found in E12.5 PSNs, confirming also our genetic tracing data in Fig. 3 indicating that competition selects for TRKCHigh/RUNX3High PSNs, the other population being eliminated.

Altogether, those results reveal an unexpected genetic variability amongst developing PSNs that distinguishes two main subgroups prior to the cell death period (Fig. 5g). They also suggest that these transcriptomic differences could represent the genetic substrates defining both the survival potential of individual neurons during competition with neighboring cells and their functional, maturation states before the neurons acquire transduction competence[40].

**RA controls RUNX3 expression in PSNs.** The postmitotic induction of RUNX3 in PSNs prompted us to examine the potential extrinsic signaling pathways regulating RUNX3 expression in presumptive PSNs before the cell death period. Interestingly, the RA-synthetizing enzyme RALDH2 is found expressed around the nascent DRG during the early phase of DRG neuron generation (E9.5–E10.0) and precedes the appearance of RUNX3/TRKC PSNs observed at E10.5 (ref. [44]). To define the contribution of RA signaling to PSNs differentiation, we first used chicken DRG, which can clearly be distinguished and dissected at the early stages of DRG formation. Similar to mouse, RALDH2 enzyme was expressed in the mesenchyme surrounding the nascent DRG following a dorso-ventral gradient and in the motor neurons (MNs) area of HHst23 chicken embryos (Fig. 6a), before RUNX3 induction[17,45,46]. HHst23 DRG explants cultured for 12 h did not express RUNX3 (Fig. 6b, Ctr), even in the presence of NT3 (data similar to Ctr), while in contrast, all-*trans* RA induced RUNX3 expression in ISL1+ neurons (Fig. 6b, RA). Interestingly, increasing concentration of RA increased the levels of RUNX3 expression per individual cells (Fig. 6c). Thus, these results show that RA can induce a dose-dependent expression of RUNX3.

To study the role of RA on Runx3 expression in PSNs in vivo, we used *Raldh2−/−* embryos treated by a short maternal RA administration (E7–E9) to allow fetal development[47], creating thus an RA deficiency before RUNX3 onset. In line with the in vitro results, a significant proportion of TRKC+ neurons failed

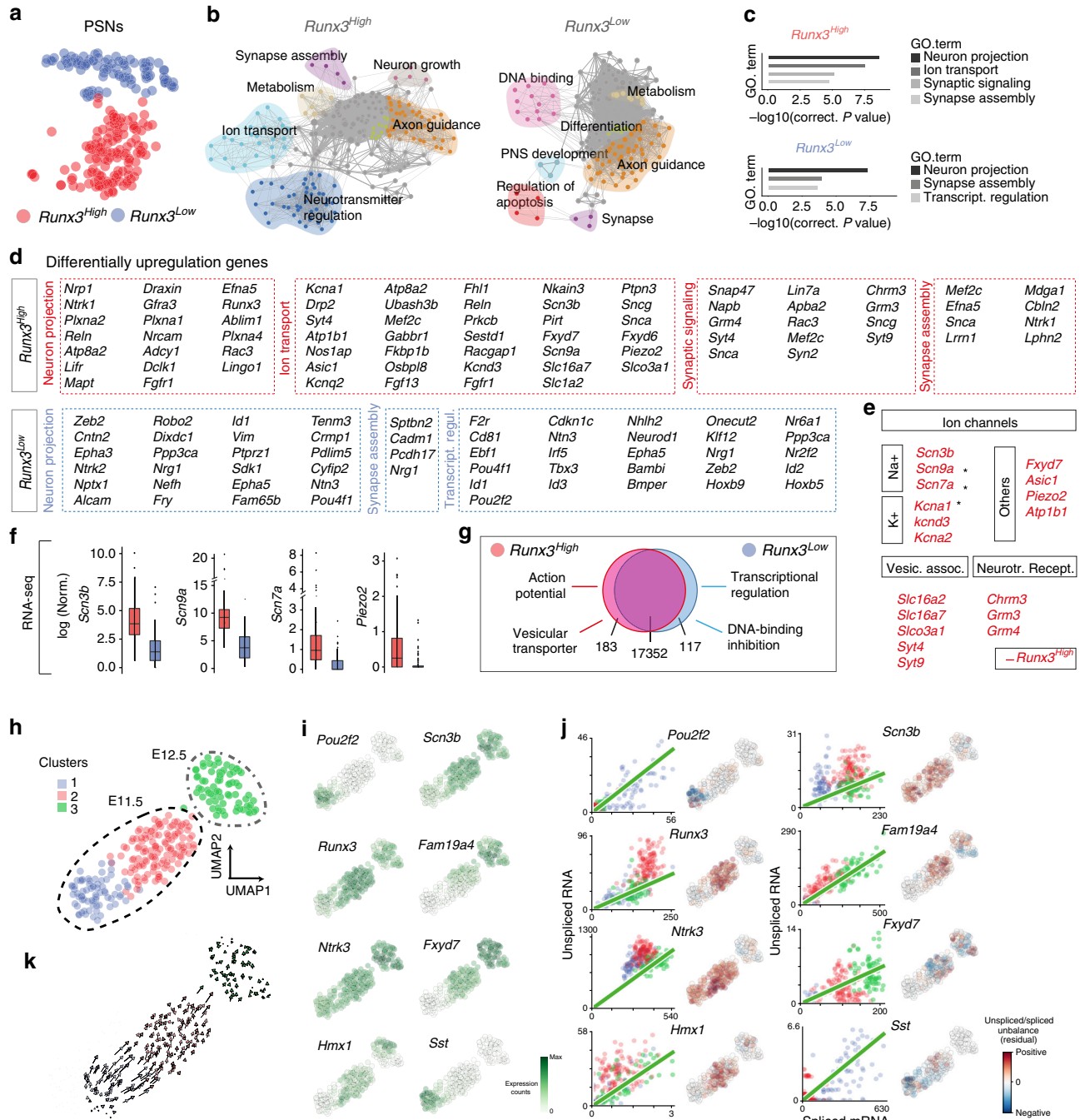

**Fig. 5** Single-cell data identifies contrasted genetic signatures of neuronal maturation amongst PSNs subpopulations. **a** *t*-SNE plots representing the two groups of PSNs at E11.5 described in Fig. 4. **b**, **c** Gene set enrichment analysis of *Runx3*^High^ and *Runx3*^Low^ neurons visualized by network. Each node represents a GO term, edges are drawn when there are shared genes between two GO terms. Interestingly, the neurotransmission category is highly represented in the *Runx3*^High^ population while apoptosis and DNA-binding are more represented in the *Runx3*^Low^ population. **c** Gene ontology analysis of the *Runx3*^High^ (top panel) and *Runx3*^Low^ (bottom panel). The graph shows most significant terms reflecting neuronal features. **d** Examples of differentially expressed genes found in *Runx3*^High^ or *Runx3*^Low^ subpopulations of PSNs. **e** Expression of select genes that encode proteins relevant to neuronal physiology (asterisks depict genes important for the generation of action potentials). Note that this category of genes is uniquely over-represented in *Runx3*^High^ PSNs. **f** Expression of genes important for the generation a mature action potentials (*Scn3b/7a/9a*) and for mechanotransduction in PSNs (see text for details). Boxplots visualize the summary of the dataset (minimum, lower quartile, median, upper quartile and maximum). **g** Summary of the differential expression profile of the two subpopulations of PSNs, with a majority of genes (17,352 genes) showing similar patterns; genes enriched in *Runx3*^High^ PSNs encode proteins associated with synaptic communication. **h** UMAP plot of PSNs single cells RNA-seq data from E11.5 and E12.5 (before and after cell death period, respectively), revealing two clusters at E11.5 (blue and red, corresponding to the clusters in **a**) and only one cluster at E12.5 (green). **i** Expression pattern using UMAP plots with specific *Runx3*^High^ and *Runx3*^Low^ marker genes shows that marker genes of E11.5 *Runx3*^High^ remain expressed at E12.5.
**j** Unspliced–spliced phase portraits (left, cells colored according to **a**) and *u* residuals (right) are shown for each gene in **i**. **k** Velocity estimates projected onto the two-dimensional UMAP plot of the dataset shown in **h**, and with a similar color code. Velocity vectors indicate positive velocity (strong directional flow) from one cell state to another and overall can be interpreted as a maturation directionality from Cluster 1 to Cluster 3. Source data are available as a Source Data file

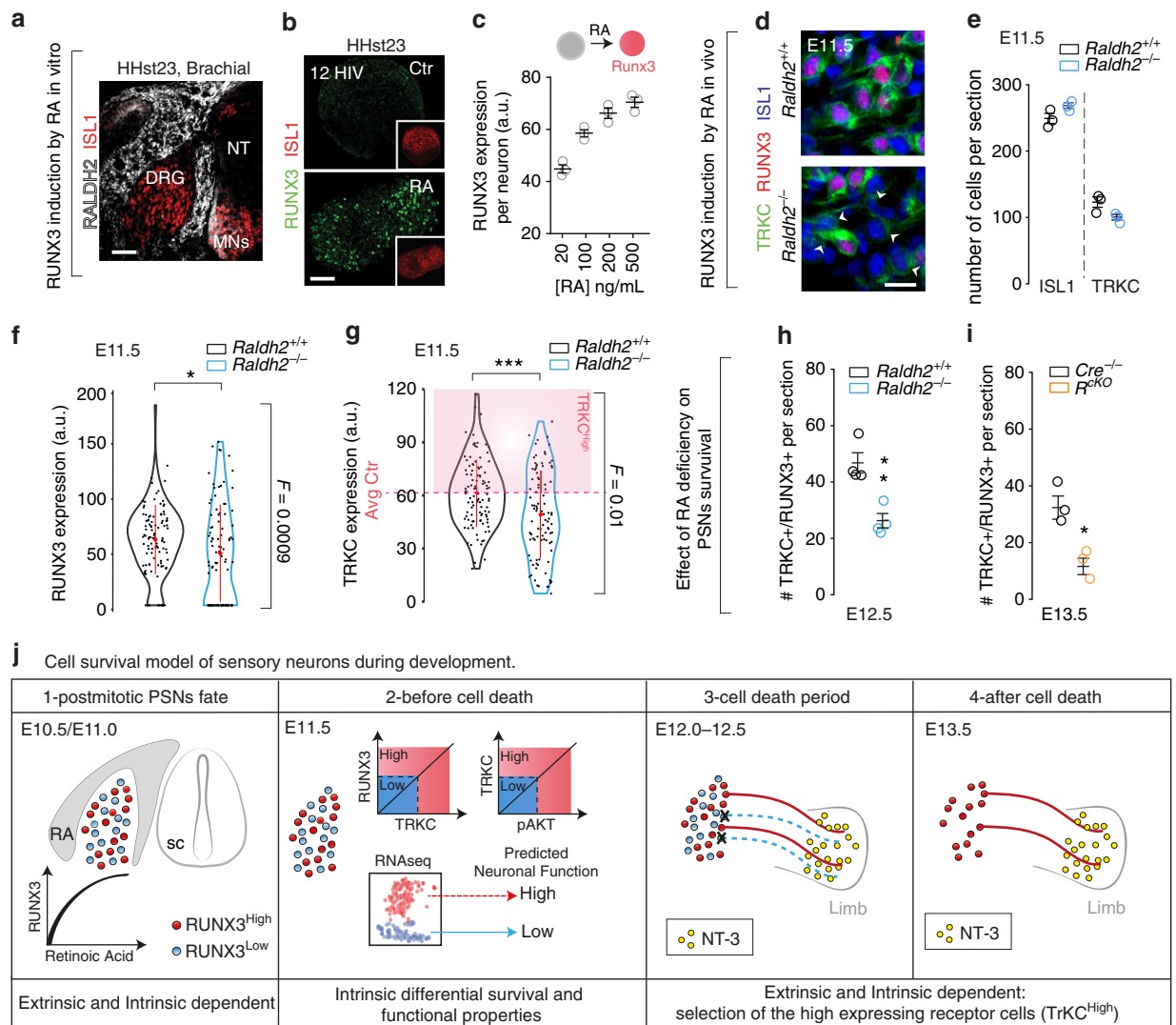

**Fig. 6** Retinoic acid signaling regulates induction of RUNX3 expression in PSNs. **a** Section of the brachial region of an HHst23 chicken embryo showing expression of the RA-synthesizing enzyme RALDH2 at the time of physiological induction of RUNX3. Scale bar: 50 μm. **b** RUNX3 expression is induced in the presence of RA (500 ng/ml) in a subset of ISL1$^+$ DRG neurons from HHst23 embryos within 12 h of treatment. Scale bar: 50 μm. **c** Concentration-dependent response of RA treatment on RUNX3 levels ($n = 3$). **d** Immunohistochemistry for ISL1, TRKC and RUNX3 on brachial DRG sections from E11.5 WT and $Raldh2^{-/-}$ mice. Scale bar: 20 μm. **e** Quantification of **d**, showing no difference in the number of ISL1 or TRKC positive neurons in $Raldh2^{-/-}$ mice ($n = 3$, unpaired Student's $t$-test). **f** Quantification of RUNX3 expression in TRKC$^+$ neurons of **d** shows an overall decrease in intensity ($^*P < 0.05$, unpaired Student's $t$-test) and a significant $F$ test ($F = 0.0009$) indicating a larger spreading of the variance between $Raldh2^{+/+}$ and $Raldh2^{-/-}$ mice ($n = 3$). **g** Quantification of TRKC expression in **d** shows significant decrease in its level of intensity per cell in the $Raldh2^{-/-}$ compared to $Raldh2^{+/+}$ mice ($^{***}P < 0.001$, unpaired Student's $t$-test). **h** Quantification of TRKC$^+$/RUNX3$^+$ DRG neurons on sections from E12.5 $Raldh2^{+/+}$ and $Raldh2^{-/-}$ mice shows a twofold decrease in the number of PSNs in the mutants ($^{**}P < 0.01$, unpaired Student's $t$-test, $n = 4$). **i** Number of PSNs is also largely decreased at E13.5 in the $Raldh1/2/3^{cKO}$ ($R^{cKO}$) mutant mice compared to the control mice $Raldh1/2/3$ ($Cre^{-/-}$) ($n = 3$, $^*P < 0.05$, unpaired Student's $t$-test). **j** Proposed model for the selection of the TRKC DRG neurons during development, in which neurons, soon after being generated, feature different functional molecular signatures that are predictive of their survival or death during the cell death period (E12–E12.5). Source data are available as a Source Data file

to induce RUNX3 expression in $Raldh2^{-/-}$ mice (29%, $n = 4$, $P = 0.005$, $t$-test, Fig. 6d, f), while the total number of neurons and TRKC$^+$ cells remained unchanged (Fig. 6e). Consistent with the role of RUNX3 on TRKC levels, presumptive PSNs that failed to acquire RUNX3 expression in $Raldh2^{-/-}$ mice showed a significantly lower expression of TRKC per cell than PSNs in $Raldh2^{+/+}$ mice (Fig. 6d, g). Notably, analysis of $Raldh2^{-/-}$ mice after the cell death period (E12.5) showed an almost twofold decrease in the number of PSNs compared to $Raldh2^{+/+}$ mice (Fig. 6h). To prevent possible compensatory mechanisms for the loss of RALDH2 by ectopic expression of RALDH1 and

RALDH3, we used a mutant mouse model in which the three members of the $Raldh$ family are deleted conditionally: $CAGG^{CreER};Raldh1^{fl/fl};Raldh2^{fl/fl};Raldh3^{fl/fl}$, hereafter named $Raldh1/2/3^{cKO}$. When induced by tamoxifen at E10, $Raldh1/2/3^{cKO}$ mice do not need RA supplementation to survive[48]. However, this timing of induction does not permit the analysis at the brachial level, where RUNX3 expression is observed from E10.5. Since RUNX3 induction follows the rostral-to-caudal gradient of normal DRG development, we characterized the phenotype at lumbar levels, where RUNX3 is first observed at E11.5. Similarly to the rescued $Raldh2^{-/-}$ mice, E13.5 $Raldh1/2/$

$3^{cKO}$ mutant mice showed a large decrease in the number of TRKC[+]/RUNX3[+] neurons (47%, Fig. 6i). Altogether, these results provide evidence that RA signaling in vivo is sufficient to induce RUNX3 in presumptive PSNs. The observation of some RUNX3 induction remaining in both mutant models could be due either to compensatory mechanisms, through activation of parallel signaling, or most likely, to an incomplete reduction of RA signaling during the critical period of RUNX3 induction using our strategies (incomplete efficiency of the recombination or of the clear-out of the RA supplement). Importantly, our results suggest a model in which the instructive role of RA on PSNs differentiation might eventually impact their differential capacities to interact with and integrate environmental cues for their survival (Fig. 6j). It is worth noting that although distinct levels of trophic factor receptors between neurons might participate in the fitness characteristic of the cell, other differentially expressed genes could be contributing to preferential cell survival.

## Discussion

The prevailing view explaining cell death mechanisms in the developing mammalian nervous system is mostly built on the principle of a one-sided control of cell death by the environment and a stochastic selection of the surviving neurons. This view has been supported indirectly by observations that experimental manipulations of neurons' environment impact their survival[2,3]. Our findings support an alternative model in which sensory neurons are endowed with distinct molecular features that are predictive of their probability to respond to environment-derived survival signals and thus, to survive the cell death period. We find that prior to cell death, sensory neurons exhibit distinct functional and survival states that are genetically encoded, involve RA signaling and RUNX3 transcriptional activity. These states are independent of signals derived from the target tissue and of NT signaling and eventually lead to the survival selection of a defined neuronal population. Thus, an early integration of cellular signaling and transcriptional programs driving cell fate decisions are coupled to a genetically distinct functional and survival fitness and thus life–death fate outcomes of neurons during competition for NTs (Fig. 6j).

Past studies on the developmental cell death of neurons in vertebrates have assumed that all TRK-positive sensory neurons are initially equivalent and that their individual probability to survive the competition period is unpredictable[4,13,14]. The selective nature of the survival-or-death choice of PSNs during development that we now reveal contrasts with this principle. Our results show that two subgroups of PSNs display distinct intrinsic capacities to compete and thus to survive prior to the cell death period. Strikingly, our data show that the molecular fitness characteristics of sensory neurons emerge very early, i.e. soon after cell cycle exit, and before the selection of nerve trajectories. Moreover, cell death of sensory neurons is a relatively fast process (~12 h) and occurs days before the nerve endings contact their final target[26]. Thus, contrary to the classic competition model, in which the size of the end organ target field would dictate that of the neuronal population by competition for target-derived trophic factors, the final set of sensory neurons, which is defined by the cell death program, would determine the number of peripheral sensory organs being innervated. In this context, a more selective mechanism of cell death might provide an advantage to guaranteeing a rapid clearance process of neurons during early development. The systems matching principle has been proposed to endow the nervous system with the capacity to adapt quantitatively to environment changes. It has remained unclear, however, how neurons are selected during this developmental period, especially if neurons had equal competitiveness (stochastic selection) and if the neural network has not fully developed yet. In our study, the distinct genetic features of the two neuronal subgroups reveal molecular differences that are associated with their competitive advantage, and hence defining a fitness attribute to survive the cell death period. The advantage of this survival selection process seems to lie within the difference in the functional state of these two populations of neurons, which co-varies with the survival advantage status of the cells. Indeed, the fittest neurons displayed a more mature functional phenotype, as illustrated by the large representation of genes involved in neurotransmission in that group. Such profile is consistent with their acquisition of mature biophysical properties and mechanosensitivity during this period, while concomitantly number of neurons with immature action potentials decreases dramatically[40]. Interestingly, voltage-gated sodium channels, whose high density is essential for the generation and propagation of action potentials, were highly enriched in the fittest neuronal subgroup, as was *Piezo2* which encodes for the major mechanotransduction channel PIEZO2 in mechanosensitive sensory neurons[41,42]. Hence, our data reveal a selection model during nervous system development where the competitive advantage of a neuron and its functional state are consubstantial elements of a general fitness profile that is established by specific transcriptional programs. The result of this is the emergence of two distinct subgroups of neurons, a more fitted and a less fitted, the former out-competing the latter, which is selectively removed during competition for survival signals. The mechanisms responsible for the differential acquisition of these molecular properties by early sensory neurons remains, however, unclear. Interestingly, those are linked to the maturation trajectory of PSNs, yet our data show that the timing of neurogenesis would not contribute significantly to these differences. Provided that RA is presumably symmetrically distributed around the ganglion, stochastic process might play a role in exposing early born neurons to various levels of RA or some other molecular signaling pathways that would regulate the levels of RUNX3 and other genes that comprise the two molecularly distinct clusters of neurons.

Thus, in addition to scaling the final number of neurons to match the size of the targets, cell death in developing sensory neurons might also participate in scaling their physiological fitness. The genetic distinction of populations of neurons with various intrinsic molecular settings and fitness observed in our study could serve as a more general mechanism preceding the selection step (through cell death), which would eventually favor the most suitable cells for the establishment of functional neural networks.

Our study also suggests similarities with the developmental cell death mechanisms in *Caenorhabditis elegans* where neuron precursors feature complex death codes that predict their survival or death[49]. A particularity of the development of *C. elegans* is, however, its invariant character, with 105 cells from the neuronal lineage being programmed to die at specific places and times[49], underlying the importance of lineage programming in this organism. In this context, it remains to be determined in vertebrates whether the predisposition of some neurons to survive or die during development is also lineage dependent or acquired postmitotically.

Sensory neurons of the trunk derive from neural crest stem cells which delaminate from the dorsal neural tube, with some migrating ventrally and settling between the neural tube and the dermomyotome to generate the DRG[50]. Later during development, these neurons will connect to peripheral organs or tissue and to specific neurons in the spinal cord[18]. In the case of PSNs, their peripheral projections reach their muscle targets by E14.0, while their central projections contact MNs in the ventral spinal cord at about E17.5[51,52]. The establishment of these networks

occurs several days after the cell death period in the DRG. How then is the development of each of these spatially disconnected structures coordinated for quantitative systems matching? These three structures (MNs, sensory neurons and skeletal muscles) develop simultaneously from adjacent precursor regions[50,53,54]. Hence, the use of a common source of signaling molecules acting locally could participate in their synchronous development. RA, which is produced by the paraxial mesoderm[44–46,55], has been shown to control the cellular specification and number of limb muscle cells and MNs[54–57]. Our results provide evidence that RA regulates the specification of PSNs cell fate, which might impact their final number. Therefore, the paraxial mesoderm, through RA production, could serve as a signaling center for the coordinated development of sensory neurons, MNs and skeletal muscles and play an important part in the establishment of a future functional sensory-motor circuit.

Finally, cell-based therapies for neuronal replacement after cell loss have great potential for treating neurological disorders[58,59]. For this, in vitro culture systems are used to derive specific neuron types from stem cells, which are then transplanted locally to disperse in the targeted neural tissue. A limitation of this therapeutic strategy is, however, to optimize the survival and the integration of the transplanted (newly specified) neurons into functional networks[59,60]. Recent studies and ours suggest a heterogeneity in newly developed neurons, which affects their fitness and would eventually impact their survival and functional integration[60,61]. Therefore, a better understanding of the distinct molecular features of individual developing neurons or stem-cell-derived neurons and their correlation with in vivo cell survival and function might provide a critical step in ensuring optimal and reproducible cell preparations for successful transplantation.

## Methods

**Animals.** Wild-type C57BL6 mice were used unless specified otherwise. $Runx3^{-/-}$, $TrkC^{CreER}$, $Raldh2^{-/-}$, $Raldh1^{fl/fl}$, $Raldh2^{fl/fl}$, $Raldh3^{fl/fl}$ and the $CAGG^{CreERTM}$ mouse strains have been described elsewhere[30,48,62–65]. $Bax^{-/-}$, $R26^{CreERT2}$, $TrkC^{-/-}$ and $Ai14$ mice have been purchased from Jackson Laboratories, and $NT3^{-/-}$ and $TrkC^{Cre}$ from MMRRC. Animals of either sex were included in this study. Animals were group-housed, with food and water ad libitum, under 12 h light–dark cycle conditions.

Fertile white Leghorn eggs were incubated at 38 °C and embryos were staged according to Hamburger–Hamilton (HH) tables.

All animal work was performed in accordance with the national guidelines and approved by the local ethics committee of Stockholm, *Stockholms Norra djurförsöksetiska nämnd*.

**Treatments.** Rescued RA animals. For the RA-rescue experiments, pregnant females were treated with RA (all-*trans*-RA, Sigma) in powdered food between E7 to E9 at a dosage of 100 µg/g of food.

Conditional KO with tamoxifen treatment. Pregnant mothers of $CAGG^{CreEM}$; $Radh1^{flox/flox}$;$Raldh2^{flox/flox}$;$Raldh3^{flox/flox}$ embryos (referred as $Raldh1/2/3^{cKO}$) were administered by gavage a dose of tamoxifen (0.1 mg per g of b.w.; Sigma) dissolved in corn oil (25 mg/ml).

Cell fate tracing experiments with 4-OHT. $R26^{tdTom}$ mice were crossed to the $TrkC^{CreER}$ and to $R26^{CreERT2}$ mice for cell fate tracing of the PSNs and for timing of early and late born neurons from the first wave of neurogenesis in the DRGs. 4-OHT induction of the $TrkC^{CreER}$ mouse line was performed by single intraperitoneal injection of either low doses (0.018 g or 0.032 g per kg bw) or higher doses (0.06 g or 0.08 per kg bw) depending on the hypothesis tested (4-OHT, Sigma, H6278) into pregnant mothers at the indicated stages. 4-OHT was dissolved in 99.5% ethanol and further diluted 1:1 in Cremophor® EL (Sigma C5135) to obtain a concentration of 20 mg/ml for storage at −20 °C. 4-OHT stock solution was further diluted 1:3 in DPBS to a working concentration of 5 mg/ml upon usage. Analysis was performed on 2–8 animals from 2–3 separate experiments.

**In ovo electroporation.** A control plasmid pCAGeGFP and siRNAs were injected in ovo into the neural tube of HHst13/14 chick embryos (plasmids concentration: 1 µg/µl, siRNA concentration: 0.07 nM, Ambion). Five pulses of 50 V/cm was performed using a square wave electroporator (BTX). Embryos were collected 3 or 5 days post electroporation and processed for immunostaining.

**In vitro cultures of DRG neurons/whole DRGs.** Dissociated DRG neurons were obtained after enzymatic digestion of brachial DRG from E11.5 embryos using trypsin (0.05% Trypsin–EDTA; Gibco). Cells were plated in 24-well plates pre-coated with poly-D-lysine (0.01%; Sigma)/laminin (10 µg/ml; R&D Systems). Cells or whole DRG were cultured in N2 medium (DMEM-F12/glutamax medium with N2 supplement; Gibco) supplemented with pen/strep, gentamicin and with NT3 (Peprotech) when specified. For in vitro RUNX3 induction in whole chicken DRGs, NT3 (10 ng/ml, R&D systems), RA (500 ng/ml, Sigma) and the pan-caspase inhibitor Q-VD-OPh (10 µM, Sigma) were used. For the survival assay using mouse E11.5 DRG neurons, the majority of neurons die during the first 24 h in the absence of NTs[66], and the addition of NT3 promotes selectively the survival of PSNs[67]. For transcripts (qPCR) measurements, whole DRGs were cultured in 2-ml open tubes in closed Petri dishes for the indicated time.

**Immunostainings and RNAscope® in situ hybridization.** Animals were collected, decapitated and fixed for 1–4 h at +4 °C (4% PFA in PBS) depending on the stage, washed in PBS, cryopreserved in 30% sucrose in PBS, embedded in OCT (Tissue-Tek) and cryosectioned at 14 µm. In situ hybridization was performed using standard RNAscope protocol (ACDBio). The RNAscope probes used in this study are Mm-*Fyxd7* and Mm-*Pou2f2* (ACDBio). Dissociated cultures were fixed for 5–20 min in cold PFA (4% in PBS). Sections or dissociated cultures were incubated for 24 h at +4 °C with primary antibodies diluted in blocking solution (2% donkey serum, 0.0125% NaN₃, 0.5% Triton X-100 in PBS). Primary antibodies used were: rabbit anti-RUNX3 (gift from Jessell TM), goat anti-TRKA (1:400, R&D Systems AF1056), goat anti-TRKB (1:500, R&D Systems AF1494), rabbit anti-TRKC (1:1000, Cell Signaling 3376), mouse anti-ISL1 (1:250, Developmental Studies Hybridoma Bank 39.4D5), goat anti-RET (1:100, R&D Systems AF482), goat anti-TRKC (1:500, R&D Systems AF1404), mouse anti-βIII-tubulin (1:1000, Promega G712A), rabbit anti-pAKT (1:100, Cell Signaling 4060S), chicken anti-RFP (1:250, Rockland 600-901-379S). After washing with PBS, Alexa Fluor secondary antibodies (Live Technology; 1:500 in blocking solution) were applied overnight (at +4 °C). Phalloidin conjugated with Alexa 488 was purchased from Life Technology. Samples were then washed in PBS and mounted in DAKO fluorescent mounting medium. Staining was documented by confocal microscopy (Zeiss LSM700) using identical settings between control and experimental images. Optical sections were 2 µm in 20× overview pictures unless specified.

**Quantifications.** For neuronal counting, soma area and intensity quantifications ImageJ software was used. Only neurons with a visible nucleus were used. Quantification of molecular markers in the dorsa root ganglia was carried out on a minimum of five DRGs sections or more randomly selected sections per animal, per condition (see figure legends for n's and genotypes). In Fig. 1a, for accurate quantification and comparison of the number of PSNs across stages, we quantified the total number of PSNs in every third sections of C5 and C7 DRGs. Also, note that for the βIII-tubulin quantification, the two outer lateral rows of cells in the different DRGs sections were not included in the analysis due to saturation of signal caused by axonal track passage that are heavily stained.

**Cell intensity quantification.** Quantification of the intensity of protein of interest from DRG sections from mice was carried out by manually drawing the cell area of interest (defining the regions of interest (ROIs)); soma, nuclei and growth cone depending on the experiment. To define RUNX3 intensity in the RUNX3$^+$/TRKC$^+$ neuron, for a given RUNX3$^+$/TRKC$^+$ cell (exp_cell), the raw integrated density of a RUNX3$^-$/TRKC$^-$ cell (control_cell) (representing the background intensity), normalized to its area was subtracted from the raw integrated density of the RUNX3-positive cell (exp_cell), normalized to its area, as follows: RUNX3 intensity (exp_cell) = [RawIntDensity (exp_cell) / Area (exp_cell)] − average [RawIntDensity (control_cell) / Area (control_cell)]. Reads were averaged by section and plotted by genotype.

**Statistics.** Data were analyzed using GraphPad Prism 5 and expressed as mean ± S.E.M., except for dot plots showing S.D. The statistical test performed is reported in the figure legend; $t$ tests were two-sided. Legend for significance: *$P \le 0.05$, **$P \le 0.01$, ***$P \le 0.001$. Data distribution was assumed to be normal. No animals or data points were excluded from the analyses. No statistical methods were used to predetermine sample size but our sample sizes are similar to those generally employed in the field. Note also that each measurement was taken from distinct samples.

**qPCR.** Brachial level DRGs were dissected from six E11.5 embryos. DRGs were arranged so that each biological replicate contained five DRG (C4–C8) from one embryo, and the same from another embryo. Experimental conditions were Zero (DRG placed directly into lysis buffer after dissection), and +NT3/−NT3 (DRG cultured in for 6 h in with or without recombinant human NT3 (Peprotech, cat. no. 450-03). DRG were cultured in DMEM:F12 (Gibco) + 10% FBS (Gibco) supplemented with 5 µM of caspase inhibitor Q-VD-OPh (Calbiochem, cat. no. 551476) in 2-ml tubes. After either dissection (Zero) or culture (+NT3/−NT3), RNA was extracted using the Qiagen RNeasy Mini Kit according to the manufacturer's instructions, including DNase I to degrade potential genomic DNA contamination. RNA quality was quantified using the

Biorad Experion Standard Sensitivity RNA Kit, which showed the RNA had RINs between 9.7 and 10. RNA was quantified using the Qubit RNA BR Assay Kit, and 50 ng of each biological replicate used for reverse transcription using Biorad iScript in a 20-µl reaction according to manufacturer's instructions. No reverse transcriptase controls were carried out at the same time by substituting water for the reverse transcriptase. Resulting cDNAs were used to analyze transcript levels using real-time PCR in a Biorad CFX96 Real-time System and using Biorad iTaq Universal SYBR® Green Supermix in 20-µl reactions for 40 cycles (95 °C 10 s, 60 °C 20 s, 72 °C 30 s followed by fluorescence measurement at 75 °C to reduce the influence of primer dimer on quantification). Primers used (5′and 3′): *B2m*, GGT CGC TTC AGT CGT CAG and TC AGT ATG TTC GGC TTC CC; *Etv1*, TAC AAG AAA CAT GGC TTG CTG AAG and CAT GAA AAG CCA AAC TTT CAG CCT; *Gapdh*, AAC TCC CAC TCT TCC ACC TTC and GAT AGG GCC TCT CTT GCT CAG; *Hprt*, GAA TCT GCA AAT ACG AGG AGT CCT and CTT TAC TAG GCA GAT GGC CAC A; *Mapk6*, TGT TGT GCA CGA GGA GGA TC and TGT GTG GCT CAG TCA TGT CG, *Ntrk3* PAN, AGT AAC CGG CTC ACC ACA CTC and AGC GGA TGT CAC AGC TGC AGT; *Ntrk3* FL, TGA TCC TCG TGG ATG GAC AG and CTT CAC TAG TAG ATT GGC TCC. *Mapk6* and *Gapdh* were included as reference genes. No template and no reverse transcriptase controls did not amplify, or were detected at least five cycles after the least strong biological sample. In addition, amplicons were sequenced to confirm specificity. Data were analyzed using qBase+ (Biogazelle, Belgium).

The truncated isoforms of *Ntrk3* cannot be uniquely measured by qPCR method. Hence, the quantity of *Ntrk3* T transcripts is deduced by subtracting the value of *Ntrk3* FL from the value of pan *Ntrk3*.

**smFISH**. Ten-micrometer-thick sections were mounted on cover glasses. Fresh sections selected for the experiment were post-fixed with 4% PFA (in PBS) for 10 min at room temperature, rinsed with PBS and permeabilized with methanol. After permeabilization, the sections were incubated for 10 min at 70 °C in Tris-EDTA pH 8.0. After washing twice with SSC 2×, the sections were incubated with hybridization buffer containing 250 nM fluorescent label probes (LGC Biosearch Technologies; see Supplementary Table 1) for 4 h at 38.5 °C. After four 20% formamide-SSC 2× washes, the slides were counterstained with Hoechst, washed with SSC 2× and mounted with Prolong Gold mounting medium (ThermoFisher Scientific). Image stacks (0.3 µm distance) were acquired using a Nikon Ti-E with motorized stage (Nikon).

The images were analyzed with a custom python script using the numpy, scipy, ndimage (http://www.scipy.org/) and scikit-image libraries[68]. Briefly, after background removal using a large kernel Gaussian filter, a Laplacian-of-Gaussian was used to enhance the RNA dots in the maximum projected image. Background objects significantly larger than the smFISH dots were removed after image thresholding and the remaining RNA dots counted in user-defined ROIs. The images were stitched, aligned and pseudocolored in Fiji. Pseudocolored images were generated by mapping the number of RNA molecules counted in each ROI to the value of a HSV color scale. For each image, a color value of zero corresponds to zero RNA molecules and the maximum color value is used to represent the highest molecular count.

Arlecchino image: after manual segmentation of the cells, the signal intensity was measured using imageJ. The signal intensity was binned in six expression classes and each class was identified with a different hue. The green color represents the lowest expression level and cyan the highest.

The truncated isoforms of *Ntrk3* cannot be uniquely labeled using the smFISH method. Hence, the number of *Ntrk3* T molecules in Supplementary Fig. 4a, b is deduced by subtracting, in individual cells, the number of *Ntrk3* FL from the number of pan *Ntrk3*.

**Single-cell isolation for single-cell analyses**. Brachial DRGs were dissected and collected in Leibovit'z L-15 medium (Life technologies) on ice. Then the DRGs were incubated in 0.05% trypsin-EDTA (1×) (Life technologies) for 5, 7, 10 and 15 min for DRGs from E9.5, E10.5, E11.5 and E12.5, respectively, at 37 °C, in a thermomixer comfort (Eppendorf) at 700 RPM. After spinning down the samples at 100 RCF for 5 min, the supernatant was removed and replaced by Leibovit'z L-15 medium (Life technologies). DRGs were physically triturated using two different sizes of pipettes previously coated with 0.2% bovine serum albumin until the solution homogenized. The cell suspension was then filtered through a 70-µm cell strainer (BD Biosciences) to remove the clusters of cells.

**Fluorescence-activated cell sorting (FACS)**. Single Tomato-positive cells were sorted by FACS into individual wells of 384-well plates containing lysis buffer. The plates containing single cells were frozen immediately on dry ice and stored at −80 °C. We used six animals (ten DRGs per animal) per experiment.

**Single-cell RNA sequencing**.

- Smart-Seq2 protocol was performed on single isolated cells by Eukaryotic Single Cell Genomics Facility at SciLifeLab, Stockholm[69]. Raw data processing:

the raw data of single-cell RNA sequencing was processed with standard pipeline by Eukaryotic Single Cell Genomics Facility at SciLifeLab, Stockholm. Most of the downstream analysis was done using R software package Seurat v2.0.

- Pre-processing (Initial filtration, normalization):

In our initial analysis of the data, we noticed contamination of non-sensory neurons in each dataset. But sensory neurons could be easily distinguished by their high expression of *Pou4f1*. Thus, we only selected cells with *Pou4f1* transcript count over 5 for further analysis.

Considering the distribution of genes identified in every cell, we selected cells with at least 4000 unique genes detected for downstream analysis. This step removed empty wells and low-quality cells. Genes expressed in less than three cells were also removed from analysis.

We used a global normalization strategy incorporated in SEURAT package to make the data comparable between different cells and between different SMART-seq2 runs. The gene expression measurements within each cell were scaled by a constant factor 10,000, then natural-log transformed. This generated a new gene expression matrix $y_{i,j} = \log(x_{i,j}/Z_j \times 10,000)$, where $x_{i,j}$ is the count of gene $i$ in cell $j$ and $Z_j$ is the total counts of all genes in cell $j$.

We calculated the dispersion (ratio of variation to mean) of genes, excluding those lowly expressed on average (normalized value <0.0125) or less variable (variation <0.5). The remaining 4008 variable genes were used for dimensional reduction. We also regressed out the cell–cell variation in gene expression driven by percent.mito (the percentage of mitochondrial gene content) using ScaleData function.

- Linear dimensional reduction:

We used principal component analysis (PCA) to reduce the dimensionality of the data. Twenty PCs were computed and first 15 PCs were determined as statistically significant through the JackStraw method. We then applied *t*-SNE to visualize the data in a two-dimensional space.

- Identify differentially expressed genes:

To identify differentially expressed genes in each cluster, we used likelihood-ratio test implemented in function FindMarkers. The parameters min.pct = 0.25 and thresh.use = 0.1 are chosen to select genes expressed in more than 25% of cells and with average expression level over 0.1 (normalized data) in the given cluster. The genes were ranked by *P*-value from low to high, with lower *P*-value indicating more differentially expressed genes.

- Boxplots: For boxplots shown in Supplementary Fig. 8, the lower and upper hinges represent the first and third quartiles, respectively, so the box spans the inter-quartile range (IQR). The horizontal line inside the box corresponds to median. The upper whiskers extend to the largest value no further than $1.5 \times IQR$ from the upper hinge, while the lower whiskers extend to the smallest value at most $1.5 \times IQR$ from the lower hinge. Dots represent the outliers beyond the range of whiskers.

**Gene Set Enrichment Analysis visualized by network**. All differentially expressed genes in *Runx3^High* and *Runx3^Low* population were used as input for Gene Set Enrichment Analysis (GSEA) using Cytoscape and its plugin Bingo. The most up-to-date gene ontology file go-basic.obo and gene association file gene_association.mgi were downloaded directly from Gene Ontology Consortium website instead of using Bingo's default ontologies or associations, which is not updated frequently. The following evidence codes were discarded from analysis: IEA (inferred from electronic annotation), ISS (inferred from sequence similarity).

Significant GO terms (*P*-value < 0.05) were used as input for Cytoscape's plugin EnrichmentMap to generate a network where mutually overlapping gene sets cluster together. Following parameters were used: *P*-value cutoff 0.001, FDR *Q*-value cutoff 0.05, similarity cutoff with Jaccard coefficient 0.25.

**RNA velocity**. We performed a pre-analysis that included the entirety of cells obtained by our sorting strategy; the aim was to isolate the *Runx3+/Ntrk3+* population in an unbiased fashion. This choice was motivated by our intention to avoid explicit filtering based on the expression of those genes, a procedure that could have excluded part of the cells of interest (because of sequencing drop-out events) or included unrelated cells (because of cross-contamination events).

As a first preprocessing step, we excluded cells with <150,000 total reads counts and red blood cells, these were easily recognized by their low number of detected genes or the extremely high hemoglobin read counts. Furthermore, we removed genes that were not reliably detected, our condition for retention was: a gene needs to have at least 38 read counts across the whole dataset and at 18 counts in at least one cell. To select the most informative genes for downstream analysis, we applied a feature selection procedure[70]. Briefly, a noise model was fit on the data and genes are ranked by the residuals to the fit, the top 1400 genes were selected. We log-transformed the data matrix and we generated a 20 nearest-neighbors graph using Euclidean distance in the space spanned by the first six principal components. This graph was, then, imputed to the Louvain community detection algorithm to cluster the cells.

For the core analysis on the *Runx3^+/Ntrk3^+* populations ($n = 274$), we used a QC and filtering procedures that identical to the one described above with the following

differences: that we selected more genes (1900 genes), more PCA components (65) and 25 nearest neighbors to construct the graph and perform Louvain clustering. These choices were aimed at extracting the most accurate representation of the dataset. To visualize the data, we used UMAP with parameters n_neighbors = 40, learning_rate = 0.6, min_dist = 0.5 and spread = 0.55. RNA velocity analysis was then performed using our tool velocyto.py[43]. We used the default parameters that we used in the original publication[43] with three differences:

- We pre-processed the data matrix only minimally (since the data is deeply sequenced and high quality), by calling velocyto.knn_imputation with parameters k = 5, b_sight = 50 and b_maxl = 20
- Only the most mature cell population (Cluster 3 in Fig. 5h) consisting entirely of cells sampled at the time point E12.5 was used to fit the degradation rate parameter (i.e. gamma). This procedure corresponds to the assumption that these cells are less rapidly changing and closer to a steady state.
- We adapted the parameter analysis to the number of cells in this dataset, in particular the estimate_transition_prob function was passed the arguments n_neighbours = 90 and calculate_grid_arrows the arguments smooth = 0.4, steps = (20, 20), n_neighbors = 40.

**Reporting summary**. Further information on research design is available in the Nature Research Reporting Summary linked to this article.

## Data availability

The accession number for the data used in this article is GEO: GSE135181. Data that support the findings of this study are also available from the corresponding authors upon reasonable request. Source data underlying Figs. 1–6 and Supplementary Figs. 2–7 are available as a Source Data file.

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

## Acknowledgements

We thank D. Ginty for the *TrkC^CreER* mice; Y. Groner and D. Levanon for the *Runx3* mice; C. Ibanez for the *R26^tdTOM* mice; P. Ernfors, H. Abdo and L. Calvo Henrique for the *R26^CreERT2;R26^tdTOM* mice; T. Jessell for the RUNX3 antibody; N. Ghyselinck and P. Dollé (IGBMC, Strasbourg) for the *Raldh* floxed mice; the CLICK imaging Facility supported by the Knut and Alice Wallenberg Foundation. We are also grateful to Prof. A. El Manira and P. Ernfors for critical reading of the manuscript. This work was supported by the Swedish Research Council (VR), the Ragnar Söderberg Foundation, Knut and Alice Wallenberg Foundation, Swedish Brain Foundation, Karolinska Institutet, the Karolinska Institutet Strategic Research program in Neuroscience (StratNeuro), the Ming Wai Lau Foundation and Åke Wiberg Foundation (F.L.); by RARENET V No. 1.7 EU Regional Development Fund, University of Strasbourg Institute of Advanced Studies (USIAS) (K.N.); and by ERC (BRAINCELL grant 261063), VR, the Wellcome Trust (grant 108726/Z/15/Z), and the EU (FP7/DDPDGENES) (S.L.). F.L. is a Ragnar Söderberg fellow in Medicine, a Wallenberg Academy Fellow in Medicine and a MWLC investigator. Open access funding provided by Karolinska Institute.

## Author contributions

S.H. and F.L. designed and supervised the experiments and either has the right to list her/himself last in bibliographic documents. Y.W., H.W., G.A. and S.H. carried out the immunostaining experiments and microscopy analysis. Y.W and S.H. performed the in vitro experiments. S.H. and H.W. performed the genetic tracing experiments. H.W. and G.L.M. performed the bioinformatics analyses, helped with related figures and wrote associated methods section. S.H. and G.L.M. conceived and supervised the RNA velocity and fate bias analyses. Y.W., A. Sh., C.P., Y.X.-F. and D.P. performed the gene expression experiments and provided help with genetic lines breeding. Y.W., H.W., G.L.M., S.H. and F.L. performed data analysis. S.C. and S.L. carried out and analyzed smFISH experiments. N.A. and I.A. performed chicken electroporation, analyzed by P.F. and Y.W. K.N., F.D.S., G.C., A. Sc. and A.M. processed and provided mouse tissues for analysis. S.H. prepared the figures. S.H. and F.L wrote the paper with input from all authors.

## Additional information

**Competing interests:** The authors declare no competing interests.

