## [Peer Review File · Nature Communications]

Reviewers' comments:

Reviewer #1 (Remarks to the Author):

This manuscript by Wang et al, addresses a very provocative and interesting problem and questions a prevailing thesis (essentially, dogma) in the field of neural development: the mechanisms that trigger the apoptotic death of subsets of neurons that die during development of the peripheral nervous system. The authors have generated a novel data set that reexamines the thesis that competition for limited amounts of neurotrophin at the target site determines which neurons die or live during development. Instead, here the thesis is that this process is dictated by early (before the period of cell death) differences in gene expression in subsets of proprioceptors that predispose one group for survival vs. death by apoptosis, in the absence of any influence of neurotrophins and/or target. In general, the data provided are sound and intriguing. There remain several unresolved questions that need to be addressed prior to endorsing this new model. Not all the data support their thesis, and not all their conclusions are supported by their data.

Specific points

1. Key point is that there are two molecularly distinct group of proprioceptors that are distinguished by levels of several genes including TrkC and Runx3. Do show difference in Runx3 RNA levels by PCR (Fig. 3e) and RNAseq (Supplementary Table) and protein, and TrkC protein, in cells differentially expressing low/high mRNAs.
2. The method for the categorization of cells by TrkC levels is not clear. Fig. 1C shows 6 levels? How were those levels parsed? By RNAseq, only 2 clusters – so what designates high vs. low TrkC expressors? The top 3 levels vs. the bottom 3 levels? A more detailed description of the methods mediating the binning/categorization of TrkC+ cells needs to be included.
3. The point that TrkC expression level is not influenced by (environmental) NT-3 is not made convincingly by the data shown in Fig. 2. The Bax^{-/-} mice are well known to be extremely abnormal so caution must be used in deducing any conclusions based on data generated with those mice. Similarly, the rest of the data to support this point comes from the Runx3^{-/-} mice, in which expression of many genes can be altered, and from (Fig.2a) in vitro data, where its well known that Trk expression is promiscuous in vitro (Yves Barde data from the early 1990s). Furthermore, the chick limb ablation studies were misinterpreted in the text here: TrkC+ proprioceptive neurons did not die in those embryos because their survival could be promoted by spinal cord-derived NT-3. Thus the most direct way to address this question would be by infusing NT-3 in utero up until E11.5 and/or use a transgenic model like that used by Davis and Albers.
4. Fig. 3 raises the question here of what exactly is being quantified: the proportion of what? The proportion of TrkC+ cells that are also TOM+? If so, this needs to be stated more clearly. If so, these TrkC+/Tom+ neurons are still a minority though, and mean that 80% of TrkC+ cells, are presumably low expressers, yet still expressed sufficient levels of TrkC to make it through the cell death period?
5. Fig. 4b, what do “MR” and “eTrkA” refer to or designate?
6. Are there still two distinct clusters of cells after the period of cell death? i.e. is the difference in gene expression solely to determine which neurons live or die, or instead, does it designate two distinct cell populations with distinct fates/identities? Simplest experiment is to stain a DRG after E12.5 and examine whether still have varying levels of TrkC expression or is the expression of TrkC uniform after E12.5? If the latter, that data would bolster the significance of the molecularly-distinct clusters of proprioceptors prior to the cell death period.
7. One concern is the fact that the Low TrkC expressing cluster are already expressing pro-

apoptosis genes, which suggests that these cells have already started down the cell death pathway so its not surprising that the cells are smaller, and express different levels of different genes. For example, cleaved caspase 3 is already late in the cell death process, so perhaps these cells already committed to dying even before E11.5? One can't conclude anything about low TrkC causing death in a cell that is already cleaved-Caspase3+. If the fundamental thesis is that lower TrkC levels predispose the cell to death, a most direct test of this thesis would be to increase expression of TrkC in vivo to test whether that increased the number of cells that survive the endogenous cell death period. As it stands now, its unsatisfying that the majority of the supporting evidence for the authors thesis is indirect.

8. Another confusing point is that the authors categorize cells by being high or low TrkC-expressing neurons, yet they are arguing that TrkC activation is not the mechanism mediating neuronal survival? If so, is it just a coincidence that the neurons that are molecularly-predisposed to survive happen to express higher TrkC levels? Some clarification of this point is required. Also why then do proprioceptors die in TrkC-/- or NT-3 knock-out mice if NT-3-TrkC interactions are not the molecular signaling pathway that determines whether a proprioceptor lives or dies during development? What is the trigger for apoptosis then if not inadequate TrkC signaling?

9. Supplementary Fig. 6c: can't make statements about cell body size since rescued neuronal biology might be abnormal.

10. Concern about fidelity of Bax-/- and Runx3-/- mice for making physiologically relevant conclusions about TrkC expression status.

11. Line 215 needs to be changed to "may correspond to the two PSN groups...."

12. Fig. 5 conclusions: One major concern is that there is considerable variability in maturation between embryos within a litter at the ages being studied here (e.g. E9.5-E12.5). For these experiments, is each embryo staged individually or are embryos from a litter lumped together? If the latter, there will be much more noise and uncertainty in the data. The authors claim that differential maturation is not the mechanism mediating the difference in cell survival, but perhaps this analysis was too subtle to be discerned in Figure 1, but not too subtle to be identified by RNAseq? An alternative hypothesis is that what is being measured here in these two clusters is that one cluster is slightly more mature than the other which would make sense since there is variability in neuronal maturation?

13. Figure 6, Line 263-265: the reduction in proprioceptors could be due to many factors (RA ko, Runx3 reduction) so to conclude that it is due to reduction in TrkC, one would need to increase TrkC and test whether that rescues these neurons from dying.

14. Finally, a more complete explanation of the role of RA on proprioceptor gene expression must be provided based on the fact that RA is presumably symmetrically distributed around the ganglion. Ultimately then some stochastic process must be invoked and/or is there some other molecular signaling pathway that regulates the levels of Runx3 and TrkC and the other genes that comprise the two molecularly distinct clusters? Again, isn't the most parsimonious explanation for the two clusters a difference in maturation? Especially since the genes that are differentially expressed include those mediating action potentials?

Reviewer #2 (Remarks to the Author):

In this manuscript, Wang et al provide evidence for neuronal competition, based on neuron "fitness," as a mechanism regulating apoptosis during the development of proprioceptive sensory neurons. For over 60 years, the neurotrophin hypothesis has been widely accepted as a theory to

explain the developmental pruning of neurons in the peripheral nervous system. The basic tenant of this hypothesis is that a given population of neurons competes for a limiting supply of neurotrophic factor produced by the target tissue; those that successfully acquire the trophic factor survive while the neurons whose axons arrive later or fail to reach the target undergo cell death. The underlying assumption of this hypothesis is that the competing neurons are largely homogeneous and that the apoptosis is stochastic. In the present study, the authors demonstrate that this assumption is not correct and that in addition to the competition for limited neurotrophin, there is also an intrinsic fitness level of the neurons, with some neurons expressing a set of genes that gives them a competitive advantage over others. They provide good evidence for this very provocative revision of the neurotrophin hypothesis; however, there are several concerns that the authors need to address to make their case more convincing.

1. The authors suggest that the levels of TrkC are critical for determining the fitness of the proprioceptive neurons, with high expressers out competing low expressors. Their evidence for this hypothesis is quite strong, particularly the elegant experiment in Fig. 3. However, in their single cell transcriptome analysis, where a TrkC reporter is used to collect the cells, they find 2 populations of neurons, based on their expression profiles, but the expression level of TrkC is not clear. The 2 groups relate to the expression of Runx3, which can regulate TrkC, but the expression of TrkC is not shown. Based on Fig. 1h, the TrkC expression is more of a continuum than 2 separate populations and in looking through the list of differentially expressed genes in Fig. 4d, TrkC (*ntrk3*) is not even listed. Did TrkC levels differ between these groups? An expression graph like those in 4g should be shown for TrkC.

2. In the graph in Fig. 2h it is not clear what exactly the y-axis is showing. The legend is not very clear. As is, the graph appears to show the % of neurons surviving relative to Runx3+ cells in 0 ng/ml NT3. If so, then why would 50ng/ml only support 4% of the Runx3+ neurons? It would be much clearer if the data was presented as relative to the survival of Runx3+ neurons in 50ng/ml, with that being 100%.

3. To further support their hypothesis that expression of TrkC determines the fitness of the neuron, the authors should increase the expression of the receptor in the neurons (e.g. using a virus) and determine if the survival in low concentrations of NT3 is higher. This experiment would be especially convincing if it could be done with expressing TrkC in Runx3^{-/-} neurons. Similarly, the TrkC^{+/-} neurons should also exhibit reduced survival. This would be a method to more directly assess the importance of TrkC expression in determining fitness. As is, the majority of the data is correlative.

Response to reviewers' comments:

Reviewer #1

We thank the reviewer for the constructive comments. Please find below a response to the criticism and an outline on the modifications introduced into the manuscript to fully address the raised concerns.

1. Key point is that there are two molecularly distinct group of proprioceptors that are distinguished by levels of several genes including TrkC and Runx3. Do show difference in Runx3 RNA levels by PCR (Fig. 3e) and RNAseq (Supplementary Table) and protein, and TrkC protein, in cells differentially expressing low/high mRNAs.

In the first version of our manuscript, we showed the relative expression of various marker genes within the two distinct subgroups of PSNs, as defined by their level of expression of RUNX3 and TRKC (Figure 4f-i). These co-labelling correlated very well with the 2 described populations of PSNs. For this type of analysis, we had to couple classic immunostainings with RNAscope method, which requires long tissue fixation for the RNAscope probes to work efficiently. This fixation is in a lot of cases incompatible with immunostaining. At the opposite, short fixations that are required for some immunostainings are incompatible with most RNAscope probes. Eventually, only few combinations are possible, after testing many conditions, and many antibodies. Nevertheless, to answer the reviewer's comment, we tried many conditions to complete the comparison of our markers' expression. While immunostaining for RUNX3 was unsuccessful following RNAscope protocol for *Runx3* and *Ntrk3* probes, we successfully labelled TRKC proteins together with its transcripts (*Ntrk3*), and with *Runx3* transcripts. We found that both *Runx3* and *Ntrk3* transcripts were expressed at high levels in cells expression high levels of TRKC (now in Fig. 4g,h). Together with the new data from Figure 4h-l and Figure 2d, and the single-cell RNAseq analysis, this provides further evidence of the two subgroups of PSNs at E11.5, as defined by TRKC^{High}/RUNX3^{High} and TRKC^{Low}/RUNX3^{Low} cells. These new data have been described in the text, and can be found in Figures 4e-g (Fig. 4f,g are shown below). Also, a new, independent analysis of the single cell measurements of PSNs using RNA velocity supports this, and is presented in new data, Figure 5h-k (used mainly in response to comments #6 and #12).

Figure legend: f,g. In vivo validation of the correlation between TRKC protein and Runx3 mRNA (f) and between TRKC protein and Ntrk3 mRNA (g).

2. The method for the categorization of cells by TrkC levels is not clear. Fig. 1C shows 6 levels? How were those levels parsed? By RNAseq, only 2 clusters – so what designates high vs. low TrkC expressors? The top 3 levels vs. the bottom 3 levels? A more detailed description of the methods mediating the binning/categorization of TrkC+ cells needs to be included.

We now have clarified in the **Materials and Methods** section the color coding of Figure 1f, and how these are connected to Figure 1g and the classification presented in Figure 1h.

3. The point that TrkC expression level is not influenced by (environmental) NT-3 is not made convincingly by the data shown in Fig. 2. The Bax^{-/-} mice are well known to be extremely abnormal so caution must be used in deducing any conclusions based on data generated with those mice. Similarly, the rest of the data to support this point comes from the Runx3^{-/-} mice, in which expression of many genes can be altered, and from (Fig.2a) in vitro data, where it is well known that Trk expression is promiscuous in vitro (Yves Barde data from the early 1990s). Furthermore, the chick limb ablation studies were misinterpreted in the text here: TrkC⁺ proprioceptive neurons did not die in those embryos because their survival could be promoted by spinal cord-derived NT-3. Thus the most direct way to address this question would be by infusing NT-3 in utero up until E11.5 and/or use a transgenic model like that used by Davis and Albers.

The Bax null mouse line has been extensively and is still extensively used in developmental neurobiology to study specification events independently of cell death events, especially in the peripheral nervous system (Fleming et al., 2016, J Neurosci; Deppmann ..., Ginty, 2008, Science; Kuruvilla..., Ginty, 2004, Cell; de Nooij ..., Jessell, 2013, Neuron; Patel ... Snider, 2000, Snider; Patel ..., Snider, Neuron, 2003; Kramer..., Arber, 2006, Neuron; Huang,..., Ginty, eLIFE, 2015), without any observation of aberrant phenotype in those tissue (other than lack of sensory neuron apoptosis) at early stages of development. Also, the control animals used in our study whenever compound mutant mice have Bax null alleles are the Bax null mice, enabling a direct comparison of the phenotypes. We therefore think that our data using Bax null background are highly relevant in our study.

We agree with the reviewer that indeed TRK expression in culture can be promiscuous, but this has been shown in dissociated cultures and after 3 days in vitro (Friedel ... Barde, PNAS, 1997). In contrast, after 3 days in vitro, explants of DRG exhibited strong differential expression of the TRK receptors depending on the selected sensory neuron type, indicating that promiscuity is only seen in dissociated neurons. Also, note the large difference of timing in vitro (72 hours in the study of Friedel et al., and only 6 hours in ours). In our study, we use DRG explants, with 6 hours of treatment, which is largely enough to upregulate *Etv1* (coding for ER81, as in Patel et al., 2003, Neuron) but not *Ntrk3*.

In regard to the last question, concerning the limb bud ablation, and referring to our previous study (Lallemend et al., 2012, EMBO J), we disagree with the interpretation of the reviewer. The claim that PSNs would survive limb bud ablation is not supported by the literature and our own unpublished data, which show an almost complete absence of PSNs following limb bud ablation after around E5 in chicken (Oakley et al., 1995, Development; Oakley et al., 1997, J Neurosci; Caldero et al., 1998, J Neurosci; as well as earlier work by V. Hamburger and R. Levi-Montalcini). Moreover, literature shows also that limb-derived NT3 promotes survival of PSNs, not from the spinal cord (see for instance Oakley et al., 1995, Development). Also, our text mentioning the limb bud ablation experiment refers only to limb-derived signal, not NT3 directly, which is instead assessed in Figure 2a-c.

4. Fig. 3 raises the question here of what exactly is being quantified: the proportion of what? The proportion of TrkC⁺ cells that are also TOM⁺? If so, this needs to be stated more clearly. If so, these TrkC⁺/Tom⁺ neurons are still a minority though, and mean that 80% of TrkC⁺ cells, are presumably low expressers, yet still expressed sufficient levels of TrkC to make it through the cell death period?

We apologize if the text was not clear in the first version of our manuscript. The text has now been changed (in the main text) to clarify the measurements and proportions observed. In response to the last question of the reviewer, our strategy using low dose of tamoxifen did show enrichment of traced cells within the TRKC^{High} population (Figure 3c,d), yet because it is a probability of recombination, and this strategy would not work with higher dose of tamoxifen, only a percentage of the TRKC^{High} population is traced. This explains the relatively low percentage of traced neurons.

5. Fig. 4b, what do “MR” and “eTrkA” refer to or designate?

The description of these populations has been added to the text of the figure legend.

6. Are there still two distinct clusters of cells after the period of cell death? i.e. is the difference in gene expression solely to determine which neurons live or die, or instead, does it designate two distinct cell populations with distinct fates/identities? Simplest experiment is to stain a DRG after E12.5 and examine whether still have varying levels of TrkC expression or is the expression of TrkC uniform after E12.5? If the latter, that data would bolster the significance of the molecularly-distinct clusters of proprioceptors prior to the cell death period.

To answer this question, we performed new single cell RNAseq analysis on E12.5 PSNs and compared newly generated transcriptomic data with single cell data of PSNs from E11.5. We then performed RNA velocity analysis to infer the developmental/maturation trajectory of our PSNs from E11.5 to E12.5. This was done in collaboration with Gioele La Manno, who published the method in Nature (La Manno et al., 2018). Similar clustering was observed at E11.5 (similar to our previous data), whereas in contrast, only one cluster of PSNs was found at E12.5, where markers found in TRKC^{High} subgroup at E11.5 were found to be expressed/maintained. This new set of experiments also answer specific question #12 (see below)

7. One concern is the fact that the Low TrkC expressing cluster are already expressing pro-apoptosis genes, which suggests that these cells have already started down the cell death pathway so its not surprising that the cells are smaller, and express different levels of different genes. For example, cleaved caspase 3 is already late in the cell death process, so perhaps these cells already committed to dying even before E11.5? One can't conclude anything about low TrkC causing death in a cell that is already

cleaved-Caspase3+. If the fundamental thesis is that lower *TrkC* levels predispose the cell to death, a most direct test of this thesis would be to increase expression of *TrkC* *in vivo* to test whether that increased the number of cells that survive the endogenous cell death period. As it stands now, its unsatisfying that the majority of the supporting evidence for the authors thesis is indirect.

The reviewer suggests that cells with smaller size would be more prone to die. We however show in supplementary Figure 2b that the TRKC levels in PSNs are not correlated with cell soma size, nor with their total mRNA content or their expression of energy/metabolism related genes (in our single cell RNAseq data).

We agree with the reviewer regarding her/his comment about cleaved-Casp3 and removed the data from the revised manuscript.

Also, the reviewer points to some apoptotic markers that are expressed in TRKC^{Low} subgroup, yet these only represent a few apoptosis-related molecules. However, the way we presented our data in our original manuscript could be confusing and lead to the interpretation of a similar selection process as seen in *C. elegans*, where some selected cells are committed to die from progenitor stage. In view of our new data using RNA velocity in figure 5, and the associated text in the main manuscript, we think that it would be more appropriate to not emphasize the few apoptotic molecules found to be enriched in TRKC^{Low} subgroup (yet, those can be still seen in the Figures and Supplementary information). Instead, and in response to the main comment here from the reviewer, we have tested the manipulation of TRKC levels and its impact on the survival probability of PSNs during the cell death period. The development of these sensory neurons is a relatively fast process, with the naturally occurring cell death starting about 2 days following PSNs emergence. Overexpression experiments is therefore not a suitable way to manipulate TRKC levels. Also, early E10.5 DRG are almost impossible to dissect out, and the success rate of early DRG neuron transfection *in vitro* is extremely low. We thus decided instead to use siRNA-based downregulation of *Ntrk3* (coding for TRKC) using chicken electroporation. Our new data show are shown in Figure 2. We demonstrate that 3 days after electroporation, at HHst26 (before cell death period), there is no loss of GFP+ PSNs in siRNA condition. In contrast, at HHst30/31, after the cell death period of PSNs, there was an 80% decrease in the number of GFP+ PSNs electroporated with *Ntrk3* siRNA, when compared with DRG sections from embryos transfected with Ctr siRNA and GFP alone (new Fig. 4i-l, see below). Hence, these findings predict a greater survival of neurons with higher TRKC levels, which would have a competitive advantage.

Figure legend: **i**, Scheme representing the *in ovo* electroporation (HHst 13/14, or E2) with either “pCAGEGFP and a negative siRNA control (Ctr siRNA)” or with “pCAGEGFP and *TrkC* siRNA (*TrkC* siRNA)”. **j**, Cross section of electroporated HHst26/27 embryos (E5) showing transfected cells in half spinal cord (SC) and in the ipsilateral DRG. **k**, Quantification at E5 (before cell death period) of TRKC intensity among transfected PSNs (RUNX3/TRKC positive), showing a significant reduction of TRKC level in *Ntrk3* siRNA condition compared to *Ctr* siRNA. **l**, Comparison before (E5) and after cell death period (E7) of the % of PSNs in transfected GFP⁺ neurons, showing significant loss of transfected PSNs at E7 in *Ntrk3* siRNA condition.

8. Another confusing point is that the authors categorize cells by being high or low *TrkC*-expressing neurons, yet they are arguing that *TrkC* activation is not the mechanism mediating neuronal survival? If

so, is it just a coincidence that the neurons that are molecularly-predisposed to survive happen to express higher TrkC levels? Some clarification of this point is required. Also why then do proprioceptors die in TrkC^{-/-} or NT-3 knock-out mice if NT-3-TrkC interactions are not the molecular signaling pathway that determines whether a proprioceptor lives or dies during development? What is the trigger for apoptosis then if not inadequate TrkC signaling?

We think here that there might be a misunderstanding. We never mentioned in the text that TRKC activation is not necessary for neuronal survival. The neurotrophin hypothesis suggesting competition between neurons for neurotrophic signaling, thus survival signal, is not in contradiction with our data, and we do not wish to make this claim. Nowhere is indicated that the selection process is independent of neurotrophin signaling. Rather, we suggest that some neurons have an advantage for competing before the cell death period starts, which challenges the “stochastic aspect” of the selection process, but not the fact that it involves a neurotrophic signaling. This is indicated at several occasions throughout the text, as for instance in the first paragraph of the Discussion: “... molecular features that are predictive of their probability to respond to environment-derived survival signals and thus, survive the cell death period” (where environment-derived signals are neurotrophins) or to mention neurotrophins directly “... life-death outcomes of neurons during competition for neurotrophins (NTs)”, where neurotrophins are clearly showed to be the survival signal for these neurons. We hope now that the reviewer does not see any intention in our manuscript to claim that neurotrophins would not play a major role in inducing survival of sensory neurons during early development.

9. *Supplementary Fig. 6c: can't make statements about cell body size since rescued neuronal biology might be abnormal.*

Cell size and hypotrophy have been largely documented in sensory neuron development to infer trophic support status (Deppmann..., Ginty, 2008, Science; Patel ... , Snider, 2000 and 2003, Neuron). Not being of a prime interest however for the paper, we removed the graph from the main figure but kept it for info in Supplementary Figure 6.

10. *Concern about fidelity of Bax^{-/-} and Runx3^{-/-} mice for making physiologically relevant conclusions about TrkC expression status.*

These comments have been answered in our responses to points #3 and #7.

11. *Line 215 needs to be changed to “may correspond to the two PSN groups....”*

We agree with the reviewer and have made changes in the text.

12. *Fig. 5 conclusions: One major concern is that there is considerable variability in maturation between embryos within a litter at the ages being studied here (e.g. E9.5-E12.5). For these experiments, is each embryo staged individually or are embryos from a litter lumped together? If the latter, there will be much more noise and uncertainty in the data. The authors claim that differential maturation is not the mechanism mediating the difference in cell survival, but perhaps this analysis was too subtle to be discerned in Figure 1, but not too subtle to be identified by RNAseq? An alternative hypothesis is that what is being measured here in these two clusters is that one cluster is slightly more mature than the other which would make sense since there is variability in neuronal maturation?*

We staged all embryos individually and only considered embryos of similar stages (as indicated in the text and figures) for comparative analysis.

The reviewer is right saying that maturation variations may still exist even though we showed that it was independent of neurons birthdate (Fig1. 1-n). This is also what we tried to emphasize in our original manuscript when mentioning in the Abstract “These molecular features are genetically encoded, representing two distinct subgroups of neurons with contrasted survival outcome and functional maturation states” or within the text of the scRNAseq experiments. Yet, it was relatively difficult to explain this in view of our experiment showing TRKC variability independent of birth date. We do think that various levels of RUNX3 might eventually lead to a greater maturation potential of the PSNs, and that the variability of RUNX3 expression triggers this differential expression and functional maturity state between cells. As described already in response to point #6, we have performed new experiments to directly assess this using RNA velocity, the most advanced method for quantitatively modeling dynamic biological processes, such as cell type specification. We thus performed new single cell RNAseq analysis on E12.5 PSNs and compared newly generated transcriptomic data with single cell data of PSNs from E11.5. We then performed RNA velocity analysis to infer the developmental, maturational trajectory of our PSNs from E11.5 to E12.5. This was done in collaboration with Gioele La Manno, who published the method in Nature (La Manno et al., 2018). Similar clustering was observed at E11.5 (similar to our previous data), whereas in contrast, only one cluster of PSNs was found at E12.5, where markers found in TRKC^{High} subgroup at E11.5 were found to be expressed/maintained. The predicted directionality of the specification/maturation of PSNs could be inferred, from RUNX3^{Low}/TRKC^{Low} to RUNX3^{High}/TRKC^{High} at E11.5, and from there to E12.5 PSNs, which form one group only. We know that PSNs later differentiate into 3 main subtypes (Ia, Ib and II afferents), but those decision events are not seen yet at E12.5, and certainly will engage further signaling from contact with their final respective targets (see also: Wu ... de Nooij JC, 2019, J Neurosci).

13. *Figure 6, Line 263-265: the reduction in proprioceptors could be due to many factors (RA ko, Runx3 reduction) so to conclude that it is due to reduction in TrkC, one would need to increase TrkC and test whether that rescues these neurons from dying.*

We agree with the reviewer and made our conclusion lighter and less conclusive on that point: “Importantly, our results suggest a model in which the instructive role of RA on PSNs differentiation might eventually impact their differential capacities to interact with and integrate environmental cues for their survival”, and in the conclusion: “Our results provide evidence that RA regulates the specification of PSNs cell fate, which might impact their final number”.

14. *Finally, a more complete explanation of the role of RA on proprioceptor gene expression must be provided based on the fact that RA is presumably symmetrically distributed around the ganglion. Ultimately then some stochastic process must be invoked and/or is there some other molecular signaling pathway that regulates the levels of Runx3 and TrkC and the other genes that comprise the two molecularly distinct clusters? Again, isn't the most parsimonious explanation for the two clusters a difference in maturation? Especially since the genes that are differentially expressed include those mediating action potentials?*

We now have discussed this further within the Discussion section (page 16, in red, see also below): “The mechanisms responsible for the differential acquisition of these molecular properties by early sensory neurons remains however unclear. Interestingly, those are linked to the maturation trajectory of PSNs, yet our data show that the timing of neurogenesis would not contribute significantly to these differences. Provided that RA is presumably symmetrically distributed around the ganglion, stochastic process might

play a role in exposing early born neurons to various levels of RA or some other molecular signaling pathways that would regulate the levels of RUNX3 and other genes that comprise the two molecularly distinct clusters of neurons.”

Reviewer #2

We thank the reviewer for the constructive criticism and provide below a response to her/his questions and an outline on the modifications introduced into the manuscript to fully address the concerns.

1. The authors suggest that the levels of *TrkC* are critical for determining the fitness of the proprioceptive neurons, with high expressers out competing low expressors. Their evidence for this hypothesis is quite strong, particularly the elegant experiment in Fig. 3. However, in their single cell transcriptome analysis, where a *TrkC* reporter is used to collect the cells, they find 2 populations of neurons, based on their expression profiles, but the expression level of *TrkC* is not clear. The 2 groups relate to the expression of *Runx3*, which can regulate *TrkC*, but the expression of *TrkC* is not shown. Based on Fig. 1h, the *TrkC* expression is more of a continuum than 2 separate populations and in looking through the list of differentially expressed genes in Fig. 4d, *TrkC* (*ntrk3*) is not even listed. Did *TrkC* levels differ between these groups? An expression graph like those in 4g should be shown for *TrkC*.

Indeed, and the data on *TRKC* expression from our scRNAseq analysis is now added in new Fig. 4e. New experiments and data have now also been added in Fig. 4f,g. Also the reviewer is right regarding the expression of *TRKC*, which appears as a continuum of levels. An RNA velocity analysis on scRNAseq data from E11.5 and E12.5 PSNs now shows the continuity of functional/maturation state between the 2 populations, and towards the E12.5 PSNs population (which appears as a single cluster). See new Fig. 5h,k (below) for new data and associated text in the Results section.

2. In the graph in Fig. 2h it is not clear what exactly the y-axis is showing. The legend is not very clear. As is, the graph appears to show the % of neurons surviving relative to *Runx3*⁺ cells in 0 ng/ml NT3. If so, then why would 50ng/ml only support 4% of the *Runx3*⁺ neurons? It would be much clearer if the data was presented as relative to the survival of *Runx3*⁺ neurons in 50ng/ml, with that being 100%.

Indeed, this is confusing and we changed the numbering to refer to 50 ng/ml.

3. To further support their hypothesis that expression of *TrkC* determines the fitness of the neuron, the authors should increase the expression of the receptor in the neurons (e.g. using a virus) and determine if the survival in low concentrations of NT3 is higher. This experiment would be especially convincing if it could be done with expressing *TrkC* in *Runx3*^{-/-} neurons. Similarly, the *TrkC*^{+/-} neurons should also exhibit reduced survival. This would be a method to more directly assess the importance of *TrkC* expression in determining fitness. As is, the majority of the data is correlative.

In response to this comment, we have tested the manipulation of TRKC levels and its impact on the survival probability of PSNs during the cell death period. The development of these sensory neurons is a relatively fast process, with the naturally occurring cell death starting about 2 days following PSNs emergence. Overexpression experiments is therefore not a suitable way to manipulate TRKC levels. Also, early E10.5 DRG are almost impossible to dissect out, and the success rate of early DRG neuron transfection in vitro is extremely low. We thus decided instead to use siRNA-based downregulation of *Ntrk3* (coding for TRKC) using chicken electroporation. Our new data show are shown in Figure 2. We demonstrate that 3 days after electroporation, at HHst26 (before cell death period), there is no loss of GFP⁺ PSNs in siRNA condition. In contrast, at HHst30/31, after the cell death period of PSNs, there was an 80% decrease in the number of GFP⁺ PSNs electroporated with *Ntrk3* siRNA, when compared with DRG sections from embryos transfected with *Ctrl* siRNA and GFP. Hence, these findings predict a greater survival of neurons with higher TRKC levels, which would have a competitive advantage.

Figure legend: **i**, Scheme representing the *in ovo* electroporation (HHst 13/14, or E2) with either “pCAGeGFP and a negative siRNA control (Ctr siRNA)” or with “pCAGeGFP and *TrkC* siRNA (*TrkC* siRNA)”. **j**, Cross section of electroporated HHst26/27 embryos (E5) showing transfected cells in half spinal cord (SC) and in the ipsilateral DRG. **k**, Quantification at E5 (before cell death period) of TRKC intensity among transfected PSNs (RUNX3/TRKC positive), showing a significant reduction of TRKC level in *Ntrk3* siRNA condition compared to *Ctrl* siRNA. **l**, Comparison before (E5) and after cell death period (E7) of the % of PSNs in transfected GFP⁺ neurons, showing significant loss of transfected PSNs at E7 in *Ntrk3* siRNA condition.

Reviewers' comments:

Reviewer #1 (Remarks to the Author):

The authors made a serious attempt to address the reviewer concerns and have improved the clarity of the text and included new supportive data. The quality of the work is excellent and the question being addressed is very interesting for the field. Many novel findings are presented here and some of them are very compelling (e.g. Fig. 3F) and support their model. However, the key experiment to directly test their model, that is directly elevating TrkC levels per neuron and track whether that increases cell survival during the death process, is still lacking and needs to be done to prevent the conclusions from being based on correlative findings. In addition, the following concerns remain:

1. Fig. 2: The images demonstrating correlation between pAKT and TrkC expression should be shown as evidence for prevalence of different and correlated expression patterns.

2. Fig. 2L: Do not understand point of this experiment - we already know from numerous studies in mouse and chick that reduction in TrkC expression reduces survival of proprioceptors. And furthermore, the data presented in this study conflict with papers from the mid-90s showing early effects of manipulating NT3 and TrkC activation in chick embryos - prior to the period of PCD.

Furthermore, as requested initially, the critical experiment is to increase levels of TrkC prior to the cell death period. I do not understand the author's response:

"the development of these sensory neurons is a relatively fast process, with the naturally occurring cell death starting about 2 days following PSNs emergence. Overexpression experiments is therefore not a suitable way to manipulate TRKC levels. Also, early E10.5 DRG are almost impossible to dissect out, and the success rate of early DRG neuron transfection in vitro is extremely low. We thus decided instead to use siRNA-based downregulation of Ntrk3 (coding for TRKC) using chicken electroporation."

Switching to the chick is smart but instead of reducing TrkC levels, you should overexpress TrkC during the same time period (with an FP-tagged virus or plasmid) that you injected the siRNAs. Then ask what % of the transfected/infected cells are TrkC+ and survive compared to a control transfected plasmid.

3. Fig. 3: Tom+ TrkC+ cells do not look brighter (increased TrkC) than Tom- TrkC+ cells.... also in C, looks like fewer data points for TOM+ PSNs. so not clear shift is real?

4. Fig. 4: Cluster gene expression is not as clear as described in the text: for example Cluster 1 expresses TrkA whereas Cluster 2 expresses TrkB yet TrkC is not even mentioned in either cluster. Isn't it formally possible that some of these other differentially-expressed genes could be contributing to differential cell fate/preferential cell survival? Also, the difference between F and G is unclear: F does not look like an antibody stain - Runx3 should be whole nucleus and TrkC should be whole membrane and cytoplasm, and golgi. In G, why not show correlation between Runx3 protein and TrkC protein? There are antibodies available to both Runx3 and TrkC.

5. Fig. 5. change in expression of TrkC does not seem that prominent?

6. Fig. 6: This result is hard to believe as TrkC is on as early as St. 20 in the chick DRG - thus Runx3 must be on even earlier. If in fact Runx3 is required for TrkC expression, there is no way Runx3 is not on at St. 23 in vivo – alternatively, this means expression has changed in vitro.

7. Title of Supp. Fig 6 should be modified – these data do not show that "Low expression of TRKC in sensory neurons is associated with low survival rate."

7. Model problem: we know that increasing target size increases the number of sensory neurons during the period of cell death. However, according to your cell selection fitness model, why would that be? Does the increase in target size increase TrkC levels per neuron or are you arguing the concentration of trophic factors would be increased with increased target and hence the less sensitive ("Low" Trk+ cells) would be able to cross a threshold for survival? But increasing target size does not mean increased concentration of neurotrophic factor per area, it just means greater overall area that can be secreting trophic factor.

Reviewer #2 (Remarks to the Author):

The authors have addressed all of the previous concerns and extensively revised their manuscript, including additional supportive data. I now support publication in the current form.

Reviewer #1:

The authors made a serious attempt to address the reviewer concerns and have improved the clarity of the text and included new supportive data. The quality of the work is excellent and the question being addressed is very interesting for the field. Many novel findings are presented here and some of them are very compelling (e.g. Fig. 3F) and support their model. However, the key experiment to directly test their model, that is directly elevating TrkC levels per neuron and track whether that increases cell survival during the death process, is still lacking and needs to be done to prevent the conclusions from being based on correlative findings. In addition, the following concerns remain:

1. Fig. 2: The images demonstrating correlation between pAKT and TrkC expression should be shown as evidence for prevalence of different and correlated expression patterns.

This is now added in Fig.2g

2. a) Fig. 2L: Do not understand point of this experiment - we already know from numerous studies in mouse and chick that reduction in TrkC expression reduces survival of proprioceptors. And furthermore, the data presented in this study conflict with papers from the mid-90s showing early effects of manipulating NT3 and TrkC activation in chick embryos - prior to the period of PCD.

b) Furthermore, as requested initially, the critical experiment is to increase levels of TrkC prior to the cell death period. I do not understand the author's response: "the development of these sensory neurons is a relatively fast process, with the naturally occurring cell death starting about 2 days following PSNs emergence. Overexpression experiments is therefore not a suitable way to manipulate TRKC levels. Also, early E10.5 DRG are almost impossible to dissect out, and the success rate of early DRG neuron transfection in vitro is extremely low. We thus decided instead to use siRNA-based downregulation of Ntrk3 (coding for TRKC) using chicken electroporation."

Switching to the chick is smart but instead of reducing TrkC levels, you should overexpress TrkC during the same time period (with an FP-tagged virus or plasmid) that you injected the siRNAs. Then ask what % of the transfected/infected cells are TrkC+ and survive compared to a control transfected plasmid."

a) What we show in the figure 2L is that using sparse transfection method to reduce the level of expression of TRKC in few PSNs, the survival of the *Ntrk3* siRNA-transfected PSNs does not change prior to the cell death period, compared to the control condition (Fig2L) while after the cell death period, their number in the *Ntrk3* siRNA condition significantly decreases comparatively to the control.

We understand the concerns of the reviewer. However, we think that our data are not conflictual with previous studies for three main reasons.

- we only reduce levels of expression of TRKC using low concentration of *Ntrk3* siRNA, but do not abolish it.

- we only target few cells amongst wildtype cells (~7% of all PSNs), avoiding indirect role when abolishing NT3-TRKC signaling in all cells. We apologize if we did not add this information in the previous version, this is now written in the main text.

- previous experiments using antibody targeting TRKC extracellular domain that likely leads to unliganded TRKC receptors, a state which is known to allow possible intracellular (pro-apoptotic) signaling without ligand, as shown by the groups of Patrick Mehlen and Y-A. Barde (dependence receptors). Here, TRKC^{Low} cells in *Ntrk3* siRNA condition still have their extracellular domain available.

As a result, in contrast to complete knockout of TRKC, we can here analyze how those TRKC^{Low} cells can now compete with the non-transfected TRKC⁺ cells for target-derived NT3, without affecting the whole population. We observed that the capacity of *Ntrk3* siRNA-transfected cells to compete is decreased compared to control siRNA-transfected cells.

The siRNA method described here allows fine reduction of TRKC levels in few cells and is therefore more relevant than forcing over-expression in a non-physiological setting.

Please find below a more elaborated explanation.

Regarding the publications of the mid-90s, we do not consider the current finding conflictual. Indeed, those studies have shown that the absence of NT3-TRKC signaling in mice and chicken embryos leads to the death of TRKC⁺ neurons, due to a lack of trophic support (Ernfors et al., Cell, 1994; Klein et al., Nature, 1994; Oakley et al., Development, 1995; Tessarollo et al., PNAS, 1997) but also to a reduction of a substantial number of non-PSNs (Gaese et al., Development, 1994; Lefcort et al., J. Neurosci., 1996; Farinas et al., 1996; Tessarollo et al., PNAS, 1997). Several studies had suggested a role for premature neuronal differentiation and/or deficit of neuronal precursor proliferation to explain this phenotype (Farinas et al., Neuron, 1996; ElShamy et al., Neuron, 1998). What was also shown is that DRG precursor cells do not express TRKC, and that TRKC is turned on in early postmitotic neurons as early as Stage 21 in chick (Wang and Scott, J. Neurosci, 2000) and E9.5-10 in mice (Ma et al., Genes Dev, 1999, and personal observation), suggesting that any effect of a lack of NT3-TRKC signaling on proliferation of precursors would be indirect (Farinas et al., Neuron, 1996 and 1998). This was later supported by the observation that lack of apoptosis signaling (using *Bax*^{-/-} mice) could rescue the number of DRG neurons in the absence of NT3 (Patel et al., 2003). Since apoptosis does not occur in wild type DRG neuron precursors (Farinas et al., Neuron, 1996; Paratore et al., Development, 2001), the lack of NT3-TRKC signaling was suggested to affect directly survival of postmitotic neurons (Patel et al., Neuron, 2003). It is possible then that early death of TRKB and TRKC neurons in the absence of NT3-TRKC signaling could indirectly decrease the proliferation rate or induce premature differentiation of neighboring cells during neurogenesis (neurogenesis waves, Ma et al., Genes Dev, 1999). In support, the loss of 80% of early differentiating neurons (from 1st wave of neurogenesis) results in a decrease of about 50% of TRKA neurons (from 2nd wave) (Hadjab et al., J Neurosci, 2013), indicating an important role of the local neuronal environment for neurogenesis in the DRG.

We think that our data therefore are not conflictual with previous work by targeting only few cells (7% of TRKC⁺ neurons) with siRNA transfection, hence avoiding possible indirect role on precursor cells and neurogenesis. To clarify this point, we have added a sentence in the result section, within the paragraph mentioning siRNA experiments, and with references to previous studies: "It also enables the targeting of only a small proportion of the neurons (~7% of PSNs), hence avoiding possible indirect role on neurogenesis as previously shown in mice and chicken embryos in the absence of NT3-TRKC signaling (Lefcort et al., J Neurosci, 1996; Farinas et al., Neuron, 1996; Elshamy et al., Neuron, 1998)".

b) We apologize if our explanation in the first revision was not clear. Our aim was to describe the limitation of our model system not being compatible with virus experiment or ex vivo experiment in mice. Instead, we used *in ovo* electroporation and are glad the reviewer finds it smart.

Following the comments from the reviewer, we think also that the title of the corresponding Result chapter is misleading and has now been re-written as "TRKC levels are intrinsically controlled and associated with **competitive advantage**", which is more appropriate. This has been corrected throughout the whole manuscript where necessary.

In addition, the reviewers suggested to test whether TRKC expression was simply correlative to or had a direct impact on cell survival in our new model, which is a very interesting question. We have successfully addressed that question by using siRNA technology. We selected this method based on knowledge of the literature, discussions with peers and the feasibility restriction of our model system. Over-riding TRKC expression in overexpression experiment would lead to levels of TRKC above physiological levels and therefore data obtained would not be conclusive on the role of TRKC in physiological conditions. Overexpression experiments are more suitable methods when applied in a gene knockout or loss-of-function context. siRNA transfection constituted the most appropriate method to modulate TRKC levels within physiological range in a small proportion of PSNs. We show that lowTRKC-expressing PSNs have less advantage to compete during competition for neurotrophic factors. We show therefore a direct impact of TRKC level on cell survival outcome in our model. To clarify this, we have added a sentence in the Result section, in Chapter “TRKC levels are intrinsically controlled and associated with competitive advantage”, at the end of the paragraph mentioning the siRNA experiment *in ovo*. We wrote:” This suggests that low TRKC-expressing PSNs are less fitted to compete for neurotrophins during the cell death period”.

In conclusion, we show in our study:

- 1) Neurons have different molecular profile before cell death period.
- 2) We showed using lineage tracing experiment that in our model system, PSNs defined by RUNX3+/TRKC+ neurons at E11.5 become only PSNs at E13.5 and not another subpopulation of sensory neurons (Fig1 and supplementary Figure1)
- 3) that neurons with higher expression of TRKC show more pro-survival signaling (Figure 2),
- 4) Higher expression of TRKC is associated with a higher probability to survive during competition for neurotrophins *in vitro* and *in vivo*, notably using genetic tracing (Figure 2 and 3).
- 5) Lowering TRKC expression in few PSNs *in ovo* reduce their competitive capacity during competition for neurotrophins
- 6) We correlate the above phenotype with the cell transcriptional profile and show that the highTRKC population is more advanced in its maturation profile and differentiate towards the E12.5 (post cell death) PSNs while the maturation state is independent of the neurons birth time (Fig1) with a state-of-the-art and unbiased analysis of their developmental trajectory using Velocity.

Together, these data demonstrate that a subgroup of sensory neurons is more competent to compete and therefore to survive the cell death period, and that the selection of the surviving neurons during competition is not stochastic at the time of competition, but determined by their gene and cell fitness identity and maturation state. We think that together, our data are sufficient to justify the conclusion and we hope that the reviewer agrees.

3. Fig. 3: Tom+ TrkC+ cells do not look brighter (increased TrkC) than Tom- TrkC+ cells.... also in C, looks like fewer data points for TOM+ PSNs. so not clear shift is real?

In the manuscript, in the paragraph “Predisposition of a distinct PSNs subpopulation to survive *in vivo*”, we describe: “we induced limited recombination in $TrkC^{CreER};R26^{tdTOM}$ mice at E11.0 with a single injection of 4-OHT (0.018g/kg) and analyzed the traced cells before (E11.5) and after the cell death period (E13.5) (Fig. 3a). At E11.5, recombination was observed in 11% of PSNs, predominantly labelling neurons with high TRKC levels (with 75% of TOM+ PSNs expressing levels of TRKC above the average) (Fig. 3b-d), supporting our strategy to increase the probability of tracing highTRKC+/highCre+ neurons by limiting the Cre activity.” Hence, our strategy was to limit recombination (only 11% of PSNs were labelled at E11.5) and doing so, preferentially label HighTRKC cells prior to the cell death period.

This is why in Figure 3C, the graph shows lower number of cells in TOM+ PSNs. The TOM+ population of PSNs is however enriched in highTRKC cells. Yet, considering that a large proportion of highTRKC PSNs has not been labelled, it is expected to see lowTRKC but also some HighTRKC PSNs that are TOM negative in the illustrative picture. Hence, in Figure 3b, amongst 9 identified PSNs, 4 of them express high levels of TRKC, 5 low levels, and the only TOM+ PSNs expresses high levels of TRKC and represents about 11% of the PSNs.

4. Fig. 4: Cluster gene expression is not as clear as described in the text: for example Cluster 1 expresses *TrkA* whereas Cluster 2 expresses *TrkB* yet *TrkC* is not even mentioned in either cluster. Isn't it formally possible that some of these other differentially-expressed genes could be contributing to differential cell fate/preferential cell survival? Also, the difference between F and G is unclear: F does not look like an antibody stain - *Runx3* should be whole nucleus and *TrkC* should be whole membrane and cytoplasm, and golgi. In G, why not show correlation between *Runx3* protein and *TrkC* protein? There are antibodies available to both *Runx3* and *TrkC*.

The single cell RNAseq data provide information on mRNA expression which, regardless of their level, will mark subgroups of cells. The level of *Ntrk1* and *Ntrk2* expression in non-TRKA and in non-TRKB populations is very low compared to what is observed in TRKA and TRKB neurons (see below figure). Also, at this stage of development (E11.5), at brachial level, it is known that less than 5% of TRKC/RUNX3 PSNs express either TRKA or TRKB protein (our study, and Kramer et al., Neuron, 2006; Bartesaghi et al., Cell Reports, 2019). Low expression of the mRNA for TRKA for instance is also seen in Ret population, see below figure, while those cells do not express TRKA protein *in vivo* (Kramer et al., Neuron, 2006; Bartesaghi et al., Cell Reports, 2019). The maintenance of *Ntrk2* transcript at low levels at this stage in lowTRKC neurons (see below figure) could be explained by the lower expression of RUNX3 in lowTRKC neurons, as RUNX3 is known to repress TRKB (Kramer et al., Neuron, 2006); yet this low level is not transduced into protein at this stage.

While there is indication from literature that those genes are not transduced into proteins, we agree with the reviewer that other differentially-expressed genes could indeed contribute to the survival potential of the two clusters of PSNs. This has now been added at the end of the chapter “Retinoic Acid controls RUNX3 expression in PSNs”, with the note: “It is worth noting that although distinct levels of trophic factor receptors between neurons might participate in the fitness characteristic of the cell, other differentially-expressed genes could be contributing to preferential cell survival.”

Regarding the staining comments, these new data refer to earlier comments from the reviewer, for validating the expression levels of the transcripts with proteins of TRKC and RUNX3 *in situ*. Hence, *Runx3* expression (in italic, transcript) in F is analyzed using RNAscope and does not represent RUNX3 protein in the nucleus. In G, we correlated mRNA for TRKC (*Ntrk3*) with TRKC protein, the correlation between RUNX3 protein and TRKC protein can be observed in Figure 2d.

5. Fig. 5. change in expression of *TrkC* does not seem that prominent?

To allow an accurate interpretation of the gene expression detected by Smartseq2 we colored dots in the tSNE to reflect the absolute count, not relative normalized expression. Therefore, the values we show always spans from 0 (white) to the max for that gene (dark green), even if the gene is always detected at high levels, rendering the difference less visible using this method.

We believe that this coloring scheme is more transparent than a relative or Z-normalized one, sometimes used in other publications (see the figure below). In fact, it does not enhance differences and does not conceal the actual data, allowing the reader to make a better "eye-ball" evaluation of the data. For example, the reader can identify the presence of a baseline level of expression, like in the case of *Ntrk3*. We show with a separate analysis the difference observed by the color gradient in the tsne plot is significant both to t-test and ANOVA and how it would have looked using some other color schemes often used in single cell RNaseq analysis.

6. Fig. 6: This result is hard to believe as *TrkC* is on as early as St. 20 in the chick DRG - thus *Runx3* must be on even earlier. If in fact *Runx3* is required for *TrkC* expression, there is no way *Runx3* is not on at St. 23 in vivo – alternatively, this means expression has changed in vitro.

We apologize if this point was not clear in our manuscript. In both mice and chick embryos, RUNX3 in PSNs starts to be expressed at least 24 hours after TRKC induction in DRG neurons. In chick DRG, TRKC is seen in neurons as early as stage 21, while RUNX3, only after stage 24 (Chen, de Nooij and Jessell, Neuron, 2006), confirming our *in vitro* data in Figure 6. In mice, TRKC is seen at E9.5, while RUNX3, only from E10.5 (Kramer et al., Neuron, 2006; Levanon et al., EMBO J, 2002). RUNX3 is thus not necessary for induction of TRKC expression in DRG neurons, confirming earlier studies (Kramer et al., 2006; Lallemand et al., EMBO J, 2012). We instead here show evidence that RUNX3 expression is necessary to set levels of TRKC expression in PSNs following RUNX3 induction in PSNs, i.e. after E10.5 (see Figure 2).

To clarify the role of RUNX3 on TRKC expression, we now have added a sentence in the “TRKC levels are intrinsically controlled and associated with survival capacity” chapter in the result section. We write: “This confirmed earlier studies on the role of RUNX3 on TRKC levels but not on its induction (Kramer et al., 2006; Lallemand et al., 2012)”.

7. Title of Supp. Fig 6 should be modified – these data do not show that “Low expression of TRKC in sensory neurons is associated with low survival rate.”

The title has now been changed to: “EC50 of NT3 for DRG neurons and PSNs soma size analysis in E13.5 *Bax* null mice”

8. Model problem: we know that increasing target size increases the number of sensory neurons during the period of cell death. However, according to your cell selection fitness model, why would that be? Does the increase in target size increase TrkC levels per neuron or are you arguing the concentration of trophic factors would be increased with increased target and hence the less sensitive (“Low” Trk+ cells) would be able to cross a threshold for survival? But increasing target size does not mean increased concentration of neurotrophic factor per area, it just means greater overall area that can be secreting trophic factor.

This is a very relevant question. Indeed implantation of a supernumerary leg in chicken embryos has been shown in past studies to rescue ~15 to 38% of the spinal cord motor neurons normally dying during the cell death period (see for instance Holliday and Hamburger, *J Comp Neurol*, 1976). Although focusing mostly on motor neurons, qualitative observations have suggested that similar rescue could be seen for sensory ganglia. At that time unfortunately, sensory subtypes could not be discerned. Therefore, if one would hypothesize that part of the PSNs population could be rescued in these experiments, we would favor the interpretation that by enlarging the target field, one decreases competition between cells for neurotrophic factors, enabling a substantial number of additional neurons to survive. This is similar to the interpretation presented by previous authors of those studies: “We interpret our data in terms of survival of motor neurons which normally would have failed in a competition at the periphery but which were sustained by the enlarged peripheral fields (Holliday and Hamburger, *J Comp Neurol*, 1976).” We therefore think that our model is not conflictual with previous findings.

Reviewer #2:

The authors have addressed all of the previous concerns and extensively revised their manuscript, including additional supportive data. I now support publication in the current form.

We are very glad that the reviewer finds that our revised manuscript has answered her/his concerns, notably about the correlative aspect of TRKC expression and survival/competitive competence of the neurons and find it now acceptable for publication in *Nature Communications*.

REVIEWERS' COMMENTS:

Reviewer #1 (Remarks to the Author):

This work is provocative, the data are sound, and will be of interest to the field. I appreciate the thoughtfulness of the authors responses and while some of my original concerns still stand, I think there are enough interesting data here to support publication in Nature Communications as they will stimulate additional exciting experiments in the field.